# Acquisition of exogenous haem is essential for tick reproduction

**Jan Perner[1,2]\*, Roman Sobotka[3], Radek Sima[1], Jitka Konvickova[1,2], Daniel Sojka[1], Pedro Lagerblad de Oliveira[4,5], Ondrej Hajdusek[1], Petr Kopacek[1]\***

[1]Institute of Parasitology, Biology Centre of the Czech Academy of Sciences, Ceske Budejovice, Czech Republic; [2]Faculty of Science, University of South Bohemia, Ceske Budejovice, Czech Republic; [3]Institute of Microbiology, Czech Academy of Sciences, Trebon, Czech Republic; [4]Instituto de Bioquímica Médica Leopoldo de Meis, Programa de Biologia Molecular e Biotecnologia, Universidade Federal do Rio de Janeiro, Rio de Janeiro, Brazil; [5]Instituto Nacional de Ciência e Tecnologia em Entomologia Molecular, Brasil, Brazil

**Abstract** Haem and iron homeostasis in most eukaryotic cells is based on a balanced flux between haem biosynthesis and haem oxygenase-mediated degradation. Unlike most eukaryotes, ticks possess an incomplete haem biosynthetic pathway and, together with other (non-haematophagous) mites, lack a gene encoding haem oxygenase. We demonstrated, by membrane feeding, that ticks do not acquire bioavailable iron from haemoglobin-derived haem. However, ticks require dietary haemoglobin as an exogenous source of haem since, feeding with haemoglobin-depleted serum led to aborted embryogenesis. Supplementation of serum with haemoglobin fully restored egg fertility. Surprisingly, haemoglobin could be completely substituted by serum proteins for the provision of amino-acids in vitellogenesis. Acquired haem is distributed by haemolymph carrier protein(s) and sequestered by vitellins in the developing oocytes. This work extends, substantially, current knowledge of haem auxotrophy in ticks and underscores the importance of haem and iron metabolism as rational targets for anti-tick interventions.

\*For correspondence: perner@paru.cas.cz (JP); kopajz@paru.cas.cz (PK)

**Competing interests:** The authors declare that no competing interests exist.

## Introduction

Haem, the heterocyclic tetrapyrrole that conjugates divalent iron, is an essential molecule for most aerobic organisms, as a prosthetic group of numerous enzymes involved in a variety of biological processes such as cellular respiration, detoxification of xenobiotics or redox homeostasis (*Furuyama et al., 2007*; *Kořený et al., 2013*). Most organisms synthesise their own haem by an evolutionarily conserved multi-enzymatic pathway occurring in the mitochondria and cytosol. Only a few haem auxotrophs lacking functional haem biosynthesis have been described to date. Among these rare organisms that are reliant on the acquisition of exogenous haem are, for instance, a protozoan parasitic apicomplexan *Babesia bovis* (*Brayton et al., 2007*), and kinetoplastid flagellates of the genus *Trypanosoma* and *Leishmania* (*Kořený et al., 2010*). Some haem auxotrophs, such as the filarial nematode parasite *Brugia malayi* (*Ghedin et al., 2007*; *Wu et al., 2009*), acquire haem from their endosymbionts, while others, such as the free-living nematode *Caenorhabditis elegans* (*Rao et al., 2005*) obtain haem from ingested bacteria. The inability to synthesise haem *de novo* was also biochemically demonstrated for the cattle tick *Rhipicephalus (Boophilus) microplus* (*Braz et al., 1999*).

In contrast to its benefits, haem is also cytotoxic, where free haem catalyses the generation of reactive oxygen species (ROS), causing cellular damage, mainly through lipid peroxidation (*Jeney et al., 2002*; *Klouche et al., 2004*; *Graca-Souza et al., 2006*). Therefore, in all living organisms, free intracellular haem has to be maintained at a low level via strictly regulated homeostasis

**eLife digest** Ticks are small blood-feeding parasites that transmit a range of diseases through their bites, including Lyme disease and encephalitis in humans. Like other blood-feeders, ticks acquire essential nutrients from their host in order to develop and reproduce.

Iron and haem (the iron-containing part of haemoglobin) are essential for the metabolism of every breathing animal on Earth. Most organisms obtain iron by degrading haem and, reciprocally, most of the iron in cells is used to make haem. However, an initial search of existing genome databases revealed that ticks lack the genes required to make the proteins that make and degrade haem.

Perner et al. wanted to find out if ticks can steal haem from the host and use it for their own development. To achieve this, Perner et al. exploited a method of tick membrane feeding that simulates natural feeding on a host by using a silicone imitation of a skin and cow smell extracts ("l´ odeur de vache"). Ticks were fed either a haemoglobin-rich (whole blood) or a haemoglobin-poor (serum) diet. This experiment revealed that ticks can develop normally without haemoglobin, but female ticks fed a haemoglobin-poor diet lay sterile eggs out of which no offspring can hatch.

Further investigation showed that haemoglobin is vitally important as a source of haem but not as a source of the amino acids needed to produce the vitellin proteins that nourish embryos. As ticks are not armed with the ability to degrade haem, they do not acquire iron from the host haem but rather from a serum transferrin, a major iron transporter protein found in mammalian blood. Further experiments revealed that ticks have evolved proteins that can transport and store haem and so make the obtained haem available across the whole tick body.

Overall, Perner et al.'s findings suggest that targeting the mechanisms by which ticks metabolise haem and iron could lead to the design of new "anti-tick" strategies.

(*Ryter and Tyrrell, 2000*; *Khan and Quigley, 2011*). This task is a critical challenge for haematophagous parasites, such as the malarial *Plasmodium*, blood flukes or triatominae insects that acquire large amounts of haem from digested haemoglobin (*Oliveira et al., 2000*; *Pagola et al., 2000*; *Paiva-Silva et al., 2006*; *Toh et al., 2010*). Maintenance of haem balance is even more demanding for ticks, as their blood meal exceeds their own weight more than one hundred times (*Sonenshine and Roe, 2014*). Despite its importance, the knowledge of haem acquisition, inter-tissue transport and further utilisation in ticks is fairly limited. Haemoglobin, the abundant source of haem for these animals, is processed intracellularly in tick gut digest cells by a network of cysteine and aspartic peptidases (*Sojka et al., 2013*). Excessive haem is detoxified by aggregation in specialised organelles termed haemosomes (*Lara et al., 2003*; *2005*) and its movement from digestive vesicles is mediated by a recently described ATP-binding cassette transporter (*Lara et al., 2015*). Only a small proportion of acquired haem is destined for systemic distribution to meet the metabolic demands of tick tissues (*Maya-Monteiro et al., 2000*).

In the present work, we have screened available tick and mite genomic databases and found that ticks have lost most genes encoding the haem biosynthetic pathway. All mites also commonly lack genes coding for haem oxygenase (HO) that catalyzes haem catabolism, raising the question of iron source for these organisms. Using in vitro membrane feeding of the hard tick *Ixodes ricinus* (*Kröber and Guerin, 2007*), the European vector of Lyme disease and tick-borne encephalitis, we performed differential feeding of females on haemoglobin-rich and haemoglobin-depleted diets. These experiments conclusively proved that ticks completely rely on the supply of exogenous haem to accomplish successful reproduction and that iron required for metabolic processes in tick tissues does not originate from haem. We propose that the unique maintenance of systemic and intracellular haem homeostasis in ticks represents a specific adaptation to their parasitic life style, and as such offers promising targets for anti-tick intervention.

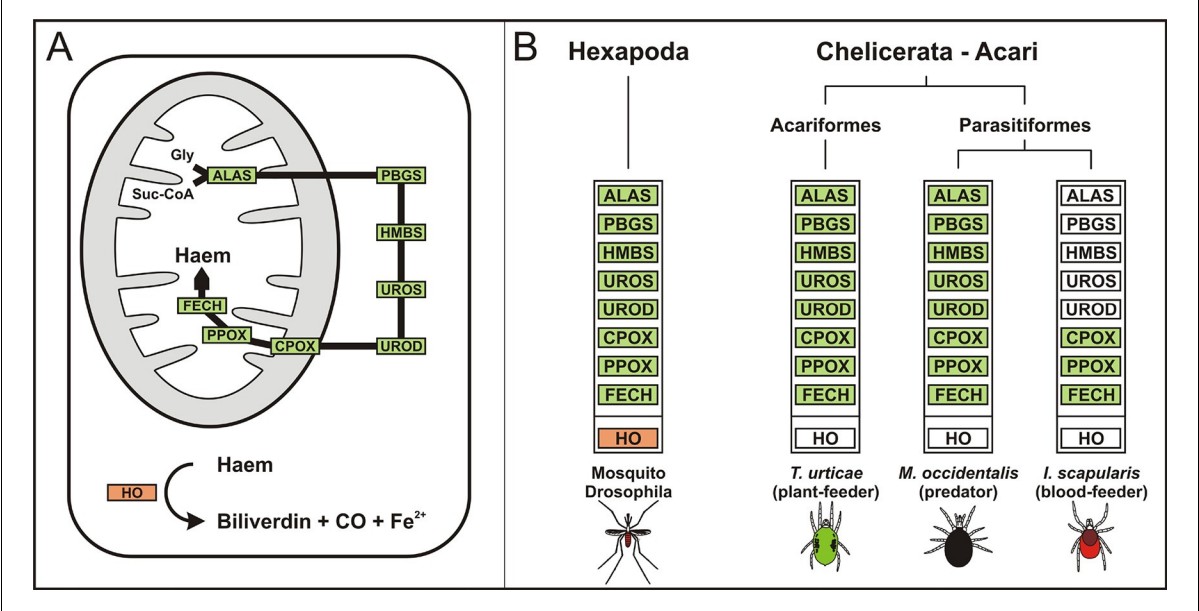

**Figure 1.** Evolution of haem biosynthetic and degradative pathways. (**A**) General scheme of haem biosynthetic and degradative pathways in the eukaryotic cell. Haem biosynthesis (upper) is a series of eight reactions beginning in the mitochondria by condensation of succinyl coenzyme A with glycine, continuing in the cell cytoplasm, and finishing in the mitochondria with the final synthesis of the haem molecule. Haem degradation (lower) is mediated by haem oxygenase in the cell cytoplasm, releasing a ferrous iron, biliverdin, and carbon monoxide. (**B**) Evolution of haem biosynthetic and degradative pathways in arthropods, according to the available genomic projects. Similarly to vertebrates, hexapods (insects) including blood feeding mosquitoes (red-coloured body), possess all enzymes for haem biosynthesis and degradation. Chelicerates lack haem oxygenase, indicating iron acquisition from sources other than haem. Plant-feeding mites (green-coloured body) of the superorder Acariformes, as well as mite-predating mites (black-coloured body) of the superorder Parasitiformes, possess a complete set of genes for haem biosynthesis. Ticks, which feed solely on blood (red-coloured body) retained only the last three enzymes (mitochondrial) of the pathway. CO - *carbon monoxide*, $Fe^{2+}$ - *ferrous iron*, Gly - *glycine*, Suc-CoA - *succinyl coenzyme A*, ALAS - *5-aminolevulinate synthase*, PBGS - *porphobilinogen synthase*, HMBS - *hydroxymethylbilane synthase*, UROS - *uroporphyrinogen synthase*, UROD - *uroporphyrinogen decarboxylase*, CPOX - *coproporphyrinogen oxidase*, PPOX - *protoporphyrinogen oxidase*, FECH - *ferrochelatase*; HO - *haem oxygenase*. Enzyme nomenclature and abbreviations according to (*Hamza and Dailey, 2012*)

The following figure supplements are available for figure 1:

**Figure supplement 1.** Phylogenetic tree of selected coproporphyrinogen-III oxidases.

**Figure supplement 2.** Phylogenetic tree of selected protoporphyrinogen oxidases.

**Figure supplement 3.** Phylogenetic tree of selected ferrochelatases.

**Figure supplement 4.** Phylogenetic tree of selected 5-aminolevulinate synthases.

**Figure supplement 5.** Phylogenetic tree of selected uroporphyrinogen decarboxylases.

## Results

### Ticks have an incomplete pathway for haem biosynthesis

The availability of the genome-wide database for the deer tick *Ixodes scapularis* (*Gulia-Nuss et al., 2016*) made it possible to analyse the overall genetic make-up for enzymes possibly participating in haem biosynthesis and compare this data with other mites and insects (Hexapoda). Complete haem biosynthetic and degradative pathways are present in insects, represented by the genomes of the fruit fly *Drosophila melanogaster* (*Adams et al., 2000*) and the blood-feeding malaria mosquito, *Anopheles gambiae* (*Holt et al., 2002*) (*Figure 1A,B*). The canonical haem biosynthetic pathway is also fully conserved in the genomes of the herbivorous mite *Tetranychus urticae*, and the predatory

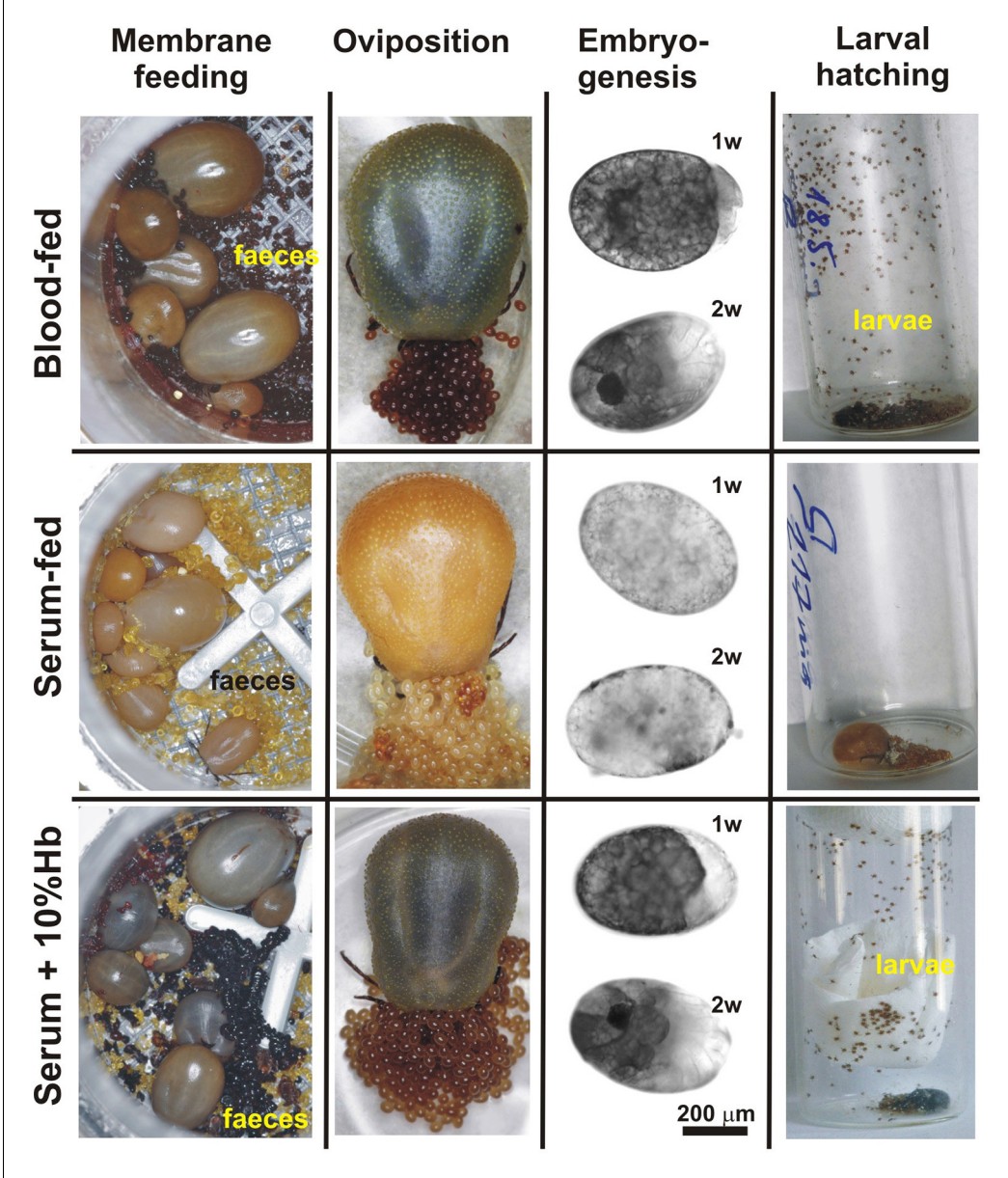

**Figure 2.** Impact of dietary haemoglobin on tick feeding, oviposition, embryogenesis, and larval hatching. (**Membrane feeding**) - membrane feeding in vitro of *Ixodes ricinus* females on whole blood (Blood-fed), serum (Serum-fed) and on serum supplemented with 10% bovine haemoglobin (Serum + 10% Hb). For dietary composition, see *Figure 2—figure supplement 1*. (**Oviposition**) – representative females laying eggs. (**Embryogenesis**) – microscopic examinations of embryonal development in eggs laid by differentially fed females; 1w, 2w – 1 week, 2 weeks after oviposition, respectively. Note, no embryos developed in eggs from serum-fed ticks, while embryogenesis was rescued in serum + 10% Hb-fed ticks. (**Larval hatching**) – Laid eggs were incubated to allow larval hatching. Note, no larvae hatched out of eggs laid by serum-fed females and the hatching was fully rescued in serum + 10% Hb-fed ticks. Similar rescue effects were also observed for ticks fed on serum supplemented with 1% and 0.1% Hb (see *Figure 2—figure supplement 2*)

The following figure supplements are available for figure 2:

**Figure supplement 1.** Diets used for tick membrane feeding and faecal examination.

**Figure supplement 2.** Rescue experiments with sub-physiological levels of haemoglobin.

mite *Metaseiulus occidentalis*, but is substantially reduced in the genome of the obligatory blood-feeding tick, *I. scapularis* (*Figure 1B*). The tick genome contains only genes encoding the last three mitochondrial enzymes of haem biosynthesis, namely, coproporphyrinogen-III oxidase (CPOX, [Vectorbase: ISCW010977], *Figure 1—figure supplement 1*), protoporphyrinogen oxidase (PPOX, [Vectorbase: ISCW023396, *Figure 1—figure supplement 2*), and ferrochelatase (FECH, [Vectorbase: ISCW016187], *Figure 1—figure supplement 3*). Corresponding orthologues could be also found in the *I. ricinus* transcriptome (*Kotsyfakis et al., 2015*) (GenBank Ac. Nos JAB79008, JAB84046 and JAB74800, respectively). Phylogenetic analyses confirmed that these genes cluster together with other Acari homologues (*Figure 1—figure supplements 1–3*, respectively). Another two gene sequences related to 5-aminolevulinate synthase (ALAS, Vectorbase: ISCW020754) and uroporphyrinogen decarboxylase (UROD, Vectorbase: ISCW020804) are clearly bacterial and most likely originate from bacterial contamination of the genomic DNA (*Figure 1—figure supplement 4* and *Figure 1—figure supplement 5*, respectively). This conclusion was further corroborated by the fact that these genes do not contain introns and are flanked by other bacterial genes in the corresponding genomic regions.

Despite an incomplete haem biosynthetic pathway, the *I. scapularis* genome contains at least 225 genes encoding a variety of enzymes utilizing haem as a cofactor, such as respiratory chain cytochromes, catalase, and a large family of cytochrome P450 genes (*Supplementary file 1*). Hence, ticks must possess efficient mechanisms for the acquisition of exogenous haem, together with its intra- and extra-cellular transport to produce endogenous haemoproteins.

## Host blood haemoglobin is expendable for tick feeding and oviposition but essential for embryonic development

In order to determine the origin of haem required for tick basal metabolism and development, we exploited an in vitro membrane feeding system developed by Kröber and Guerin (*Kröber and Guerin, 2007*). We fed *I. ricinus* females with whole blood (BF ticks), and, in parallel, with haemoglobin-free serum (SF ticks) (*Figure 2* and *Figure 2—figure supplement 1*). Serum-fed ticks were capable of fully engorging and laying eggs similar to BF ticks (*Figure 2*). However, striking differences were observed in embryonic development and larval hatching. Embryos in eggs laid by BF females developed normally as described for naturally-fed ticks (*Santos et al., 2013*) and gave rise to living larvae (*Figure 2*). In contrast, no embryonic development was observed in colourless eggs laid by SF ticks, and accordingly, no larvae hatched from these eggs (*Figure 2*). To prove that haemoglobin alone, and no other component of red blood cells, is required for successful tick development, a rescue experiment was performed. From the fifth day of membrane feeding (prior to the females commencing the rapid engorgement phase), the serum diet was supplemented with 10%, 1%, or 0.1% pure bovine haemoglobin and ticks were allowed to complete feeding (S+Hb-F ticks). The presence of haemoglobin in the diet rescued the competence of embryos to develop normally and the number of larvae hatching from eggs laid by S+Hb-F ticks was comparable with BF ticks (*Figure 2*, bottom panels). The same rescue effect was observed for ticks fed on 1% and 0.1% haemoglobin (*Figure 2—figure supplement 2*) demonstrating that as little as one hundredth of the physiological concentration of haemoglobin in the diet is sufficient to maintain tick reproduction.

## Haemoglobin is an indispensable source of haem, a replaceable source of amino acids, but not a source of iron for ticks

After a blood meal, the physiology of an adult female tick is dominated by its reproductive effort as up to half of the weight of a fully engorged female is used in the production of thousands of eggs (*Sonenshine and Roe, 2014*). To disclose the importance of haemoglobin in tick reproduction, we first determined haem levels in eggs obtained from both BF and SF ticks. The concentration of haem *b* (the form of haem present in haemoglobin) was determined by reverse-phase HPLC (*Figure 3A* and *Figure 3—figure supplement 1*). Eggs laid by BF ticks contained $669 \pm 45$ pmol haem *b*/mg eggs, whereas eggs laid by SF ticks contained virtually no haem (only $3 \pm 1.6$ pmol haem *b*/mg eggs). Eggs from the rescue experiment (S+Hb-F ticks) contained only slightly decreased haem levels ($508 \pm 79$ pmol haem *b*/mg eggs) compared to BF ticks. Eggs from ticks fed with sub-physiological levels (1% and 0.1%) of haemoglobin contained gradually decreasing haem levels ($471 \pm 17$ and

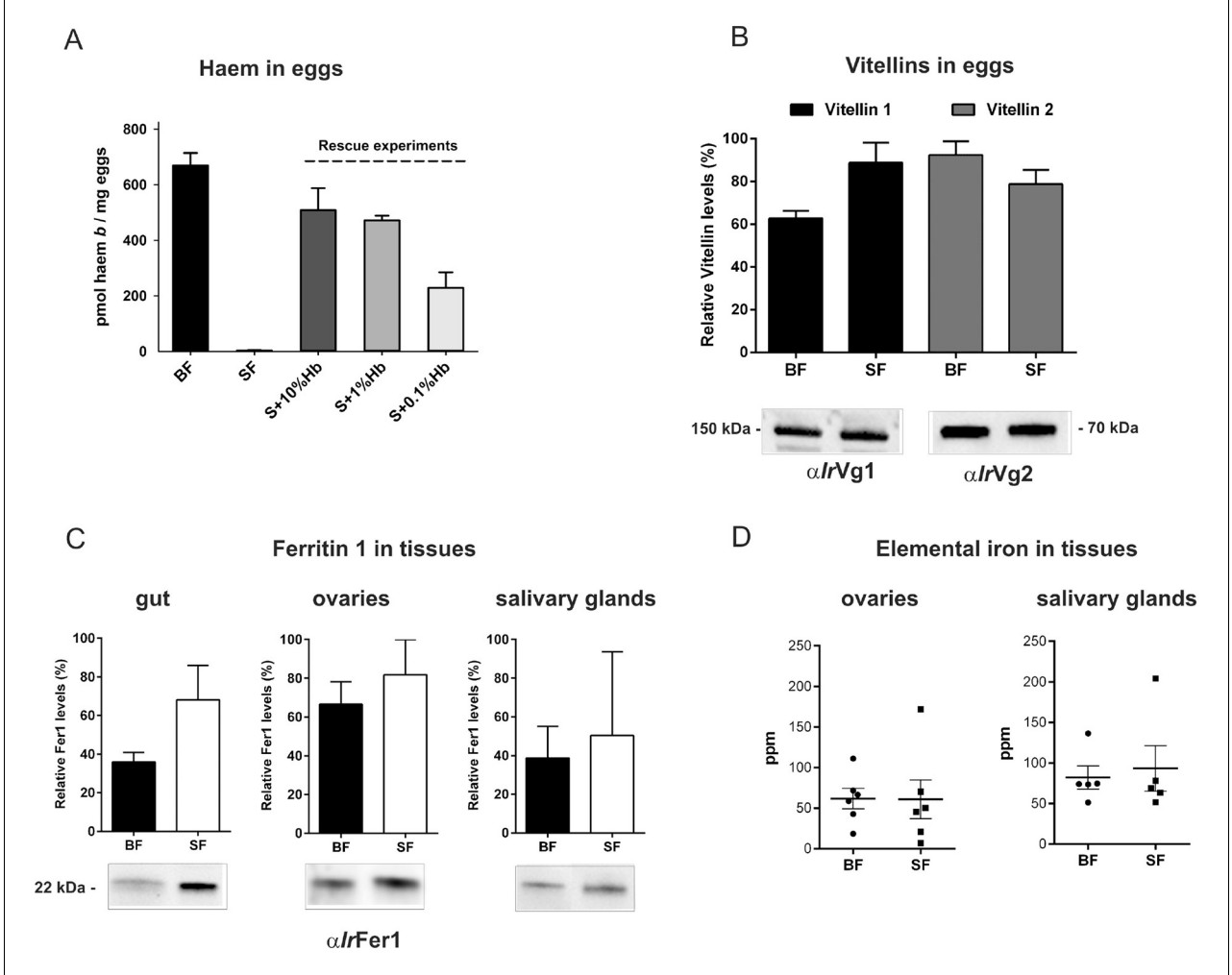

**Figure 3.** Determination of haemoglobin-derived nutrients in ticks (haem, amino acids, iron). (A) Levels of haem $b$ were determined by HPLC in egg homogenates from ticks fed on whole blood (BF) serum (SF), and serum supplemented with 10%, 1% or 0.1% bovine haemoglobin (S+10%Hb, S+1%Hb and S+0.1% Hb, respectively; rescue experiments). Data (mean values ± SEM) were acquired from homogenates of three independent clutches of eggs. Representative chromatograms detecting haem $b$ in egg homogenates are shown for BF ticks, SF ticks, and S+10% Hb - fed ticks, see ***Figure 3—figure supplement 1***. (B) Quantitative Western blot analyses detecting levels of vitellin 1 and vitellin 2 in egg homogenates using antibodies raised against vitellin precursors - vitellogenins (*Ir*Vg1, *Ir*Vg2). Bar charts depict the mean levels ± SEM of the particular vitellin in the egg homogenates from three different clutches of BF ticks or SF ticks (see ***Figure 3—figure supplement 2***). Representative Western blot detection is shown below the bar chart. (C) Quantitative Western blot analyses detecting ferritin1 (*Ir*Fer1) in the gut, ovary, and salivary gland homogenates from BF and SF ticks. Bar charts depict the mean ± SEM levels of *Ir*Fer1 in the tissue homogenates prepared from three independent tissue pools (see also ***Figure 3—figure supplement 2***). Representative Western blot detections for guts, ovaries and salivary glands are shown below the bar charts. (D) GF-AAS elemental analysis of iron in ovaries and salivary glands pools. Each data point represents a pool of five tissues dissected from BF and SF partially engorged ticks (fed for 6 days). Iron content is expressed in ppm (ng Fe per mg of dry tissue). Main and error bars indicate group means and SEM, respectively.

The following figure supplements are available for figure 3:

**Figure supplement 1.** HPLC analysis of haem b in tick egg homogenates.

**Figure supplement 2.** Full appearance of SDS-PAGE and Western blot analyses shown in the ***Figure 3***.

**Figure supplement 3.** Detection of biliverdin IX derivatives in *Ixodes ricinus* and *Aedes aegypti*.

229 ± 97 pmol haem *b* /mg eggs, respectively) (*Figure 3A*), but were still capable completing development and producing viable larvae (*Figure 2—figure supplement 2*).

Vitellins, the major tick egg yolk proteins, account for more than 90% of the protein content of a mature egg (*James and Oliver, 1997*; *Logullo et al., 2002*). In contrast to haem concentrations, no apparent differences were observed in vitellin levels in eggs from BF and SF ticks, as determined by quantitative Western blot analysis (*Figure 3B*) with specific antibodies raised against recombinant vitellin precursors, *I. ricinus* vitellogenin 1 (*Ir*Vg1) and vitellogenin 2 (*Ir*Vg2) (*Supplementary file 2*; *Figure 3—figure supplement 2*). This result implies that haemoglobin is replaceable by serum proteins as a nutritional source of amino acids needed for vitellogenesis.

Genome-wide analyses of *I. scapularis* and other mites revealed a common unique feature; the gene encoding haem oxygenase (HO) is missing, pointing to a lack of enzymatic degradation of haem in these Acari representatives (*Figure 1B*). HO-mediated haem degradation results in the equimolar release of iron and the linear tetrapyrrole product, biliverdin (*Khan and Quigley, 2011*). Gut homogenates from fully engorged *I. ricinus* females were analysed by HPLC for the presence of biliverdin IX (*Figure 3—figure supplement 3*). With the detection limit as low as 5 pmol, no trace of biliverdin IX or modified biliverdin showing a bilin-like light absorbance near 660nm was detected in *I. ricinus* gut homogenates. In contrast to ticks, the presence of biglutamyl biliverdin IX in whole body extracts of the blood-fed mosquito, *Aedes aegypti* (*Pereira et al., 2007*), was confirmed by our method exploiting diode-array detection (*Figure 3—figure supplement 3*). The lack of HO thus poses a question of the iron source for ticks. Iron availability in tick tissues was examined using two independent methods: (i) The presence of iron was indirectly tested by monitoring the levels of intracellular Ferritin 1 (*Ir*Fer1). Under iron deficiency, the translation of *ir-fer1*mRNA is suppressed by binding of the iron regulatory protein (IRP1) to its 5'-located iron-responsive element, whereas at high iron levels, the proteosynthesis of *Ir*Fer1 is up-regulated (*Kopáček et al., 2003*; *Hajdusek et al., 2009*). Homogenates of guts, ovaries, and salivary glands were analysed by quantitative Western blotting using *Ir*Fer1-specific antibody (*Figure 3C* and *Figure 3—figure supplement 2*). *Ir*Fer1 levels were lower in guts and about equal in ovaries and salivary glands of BF compared to SF ticks (*Figure 3C* and *Figure 3—figure supplement 2*); (ii) The elemental iron concentration in tick tissues was determined directly by graphite furnace atomic absorption spectrometry (GF-AAS). As this method is not able to distinguish between iron of haem and non-haem origins, only salivary glands and ovaries dissected from partially engorged BF and SF ticks were used for the analysis to avoid distortions caused by the presence of haemoglobin in the samples. Despite large variations within individual biological replicates, the average iron concentration in either tissue was independent of haemoglobin in the tick diet (*Figure 3D*). These results conclusively proved that the bioavailable iron in tick tissues originates from host serum components rather than from haemoglobin-derived haem.

## Haemoglobin-derived haem is transported from the gut to the ovaries

Guts dissected from partially-engorged *I. ricinus* females, and ovaries dissected 6 days after detachment (AD) from both BF and SF ticks displayed similar overall morphologies, except for colour (*Figure 4*). Accordingly, haem-containing haemosomes were not observed in the digest cells from SF ticks (*Figure 4*). Haemolymph collected from BF ticks displayed a typical haem light absorbance maximum (Soret peak) around 400 nm, which is not present in haemolymph from SF ticks (*Figure 5A*). This observation demonstrates that haem present in the haemolymph of fully engorged females originated only from the blood meal of adults, and not from previous feeding at the nymphal stage. We estimate that out of approximately 10 μmol of total haem acquired from a tick blood meal, only about 100 nmol (~1%) needs to be transported to the ovaries within a period of several days.

## *Ir*CP3 is the major haem-binding protein in *I. ricinus* haemolymph

Haem inter-tissue distribution and storage is facilitated by haem-binding protein(s). In the cattle tick *R. microplus*, the most abundant haemolymph protein, named HeLp, was reported to bind haem in the haemocoel (*Maya-Monteiro et al., 2000*). The genome of *I. scapularis* contains at least five genes related to HeLp, annotated as carrier proteins (*cp1–5*). In *I. ricinus,* we identified and sequenced the *cp3* orthologue, further referred to as *ir-cp3* (GenBank KP663716). Expression

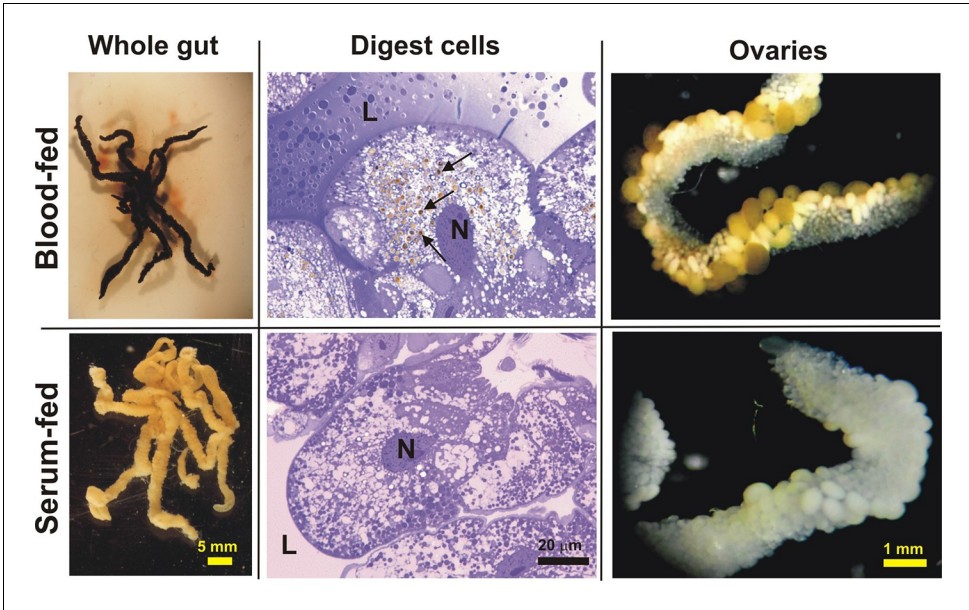

**Figure 4.** Appearance of the tick gut, digest cells, and ovaries from blood- and serum-fed ticks. Whole guts from blood-fed (BF) and serum-fed (SF) partially engorged females (fed for 6 days) were dissected and semi-thin sections of digest cells were prepared and stained with toluidine blue. L - lumen; N - nucleus; arrows point to developing haemosomes that were present only in digest cells of BF ticks. Ovaries were dissected from BF and SF fully engorged females 6 days after detachment from the membrane.

profiling over *I. ricinus* developmental stages and tissues revealed that *ir-cp3* mRNA was consistently up-regulated by blood-feeding and was predominantly expressed in the trachea-fat body complex and, to a lesser extent, in salivary glands and ovaries of adult females (*Figure 5—figure supplement 1*). SDS PAGE and Western blot analysis revealed that *Ir*CP3 was most abundant in tick haemolymph (*Figure 5—figure supplement 1*), where its levels were not affected by the presence or absence of haemoglobin in the tick diet (*Figure 5B,C*). Native pore-limit PAGE, followed by detection of haem via its peroxidase activity with 3,3'-diminobenzidine (DAB), showed that haem was associated with the ~ 300 kDa band of *Ir*CP3 only in the haemolymph from BF ticks (*Figure 5C*, DAB panel). RNAi-mediated silencing of *ir-cp3* in *I. ricinus* females (*ir-cp3* KD) resulted in the disappearance of the haem Soret peak (*Figure 5D*), a substantial (~80%) reduction in *Ir*CP3 levels on SDS PAGE (*Figure 5E*), and the absence of *Ir*CP3-associated DAB stained haem on the native gel (*Figure 5F*). These results collectively demonstrate that *Ir*CP3 is the major haem-binding protein in *I. ricinus* haemolymph.

## Vitellins are the major haem-binding protein in *I. ricinus* ovaries

Extracts from *I. ricinus* ovaries were colourless until the 3rd day after detachment (AD) from the host, and then the Soret peak absorbance gradually increased, indicating an increase in haem concentration up to 8 days AD (*Figure 6A*). SDS PAGE and Western blot analysis of ovary homogenates revealed that levels of *Ir*Vg1- and *Ir*Vg2-derived proteolytic products gradually increased after tick detachment whereas *Ir*CP3 remained constant (*Figure 6—figure supplement 1*). Native pore-limit PAGE followed by DAB-based haem co-detection and Western blot analyses confirmed that the appearance of haem in tick ovaries was coincident with the occurrence of vitellins (*Figure 6B*). *I. ricinus* vitellogenin genes (*ir-vg1* and *ir-vg2*) are exclusively expressed in fully engorged females, predominantly in the gut, salivary glands and trachea-fat body complex, but not in the ovaries (*Figure 6—figure supplement 2*). As vitellins are predominantly found in ovaries, their precursors (vitellogenins) must be transported from their site of synthesis to the ovaries.

RNAi-mediated silencing of *ir-vg1* and *ir-vg2* resulted in a substantial decrease in mRNA levels of both vitellogenin genes in gut tissues, and the same dual silencing effect was also observed at the protein level for *Ir*Vg1 and *Ir*Vg2 in tick ovary homogenates (*Figure 6—figure supplement 3*). This

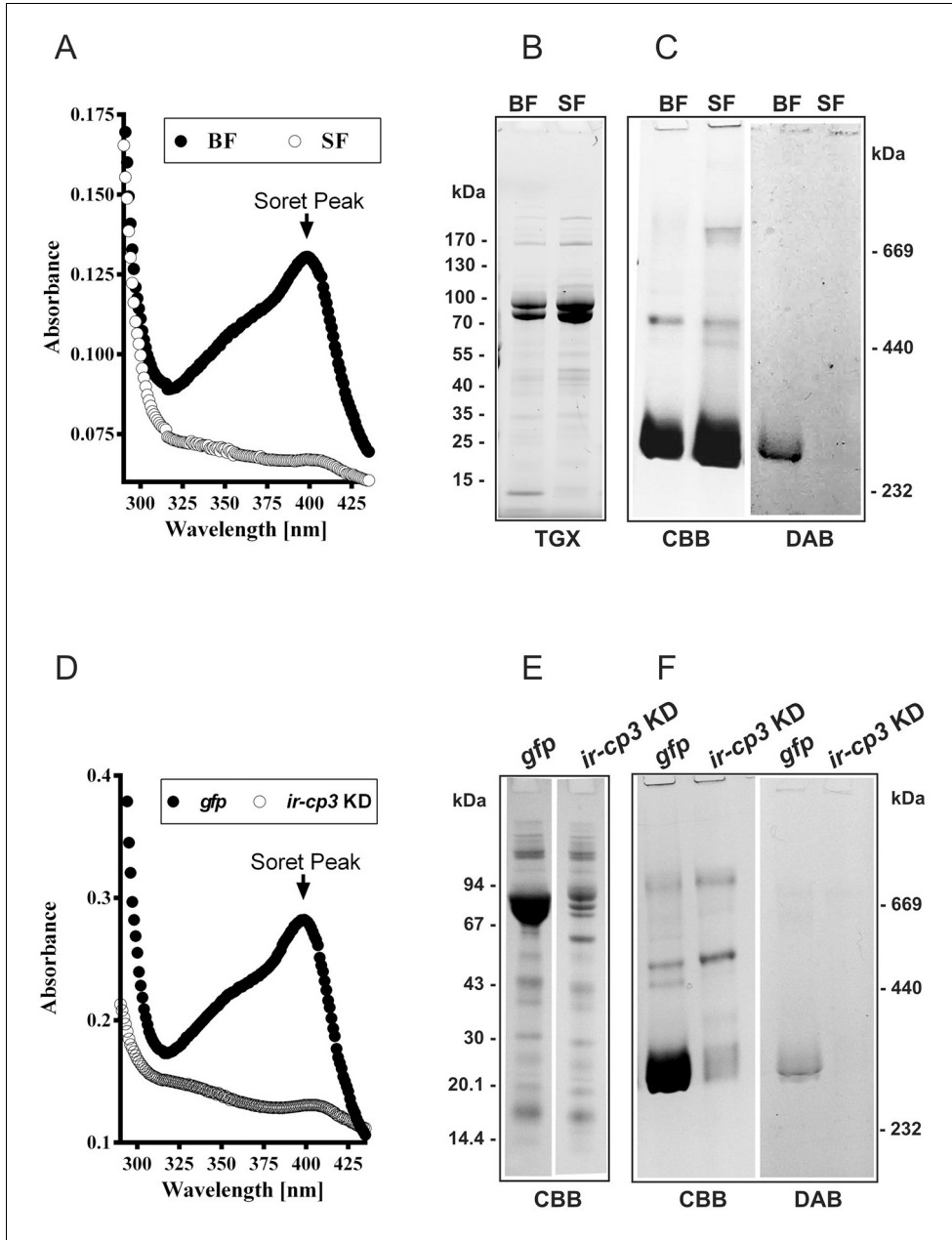

**Figure 5.** *Ir*CP3 is the major haem-binding protein in *I. ricinus* haemolymph. (**A-C**) *Ir*-CP3 and haem levels in haemolymph collected from blood-fed (BF) and serum-fed (SF) partially engorged females. (**A**) Absorbance spectra of haemolymph samples from BF and SF females. (**B**) SDS-PAGE of haemolymph samples from BF and SF ticks. Protein profiles were visualized using the TGX Stain-Free technology (TGX). (**C**) Native pore-limit PAGE of heamolymph proteins stained with Coomassie (CBB) and specific co-detection of haem using peroxidase reaction with 3,3′-diaminobenzidine (DAB). (**D-F**) Effect of RNAi-mediated silencing of *ir-cp3* on the *Ir*-CP3 and haem levels in tick haemolymph. Unfed *I. ricinus* females were injected with *gfp* dsRNA (gfp, control group) or with *ir-cp3* dsRNA (*ir-cp3* KD group) and ticks were allowed to feed naturally on guinea pigs until partial engorgement (fed for 6 days). (**D**) Absorbance spectra of haemolymph samples from from g*fp* control and *ir-cp3* KD silenced ticks. (**E**) SDS-PAGE of haemolymph proteins (10 μl, 1:20 dilution) collected from g*fp* control and *ir-cp3* KD ticks. Protein profiles were stained with Coomassie (CBB). (**F**) Native pore-limit PAGE of heamolymph proteins from g*fp* control and *ir-cp3* KD ticks. Protein profiles were stained with Coomassie (CBB) and haem was co-detected using DAB.

The following figure supplement is available for figure 5:

**Figure supplement 1.** Stage and tissue expression of *I. ricinus* haemolymph carrier protein (IrCP3).

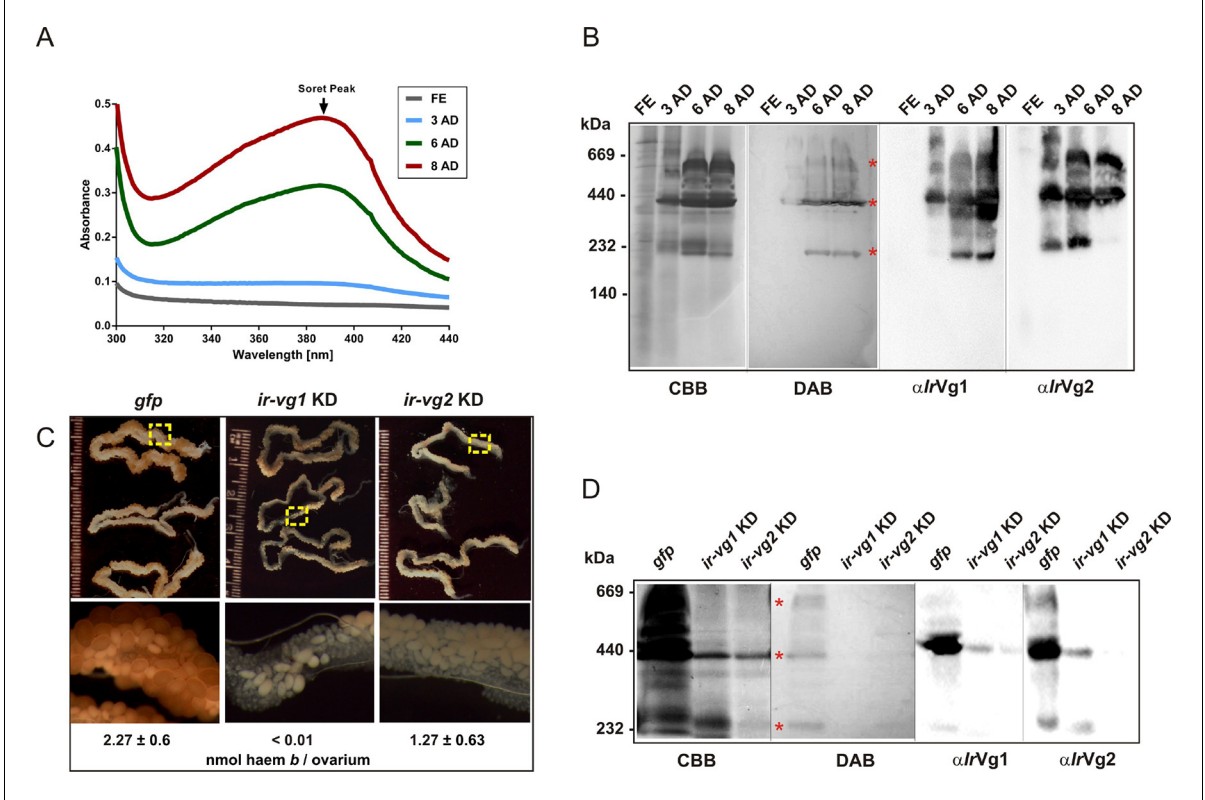

**Figure 6.** Vitellins are the major haem-binding proteins in tick ovaries. (A-B) Haem accumulation in tick ovaries occur concurrently with the appearance of vitellins. Ovaries were dissected from *I. ricinus* females at subsequent time-points after detachment (AD) from the host: FE - fully-engorged; 3 AD, 6 AD, 8 AD - 3, 6, and 8 days AD, respectively. (A) Absorbance spectra of ovaries homogenates show gradually increasing Soret peak following the 3rd day AD. (B) Native pore-limit PAGE of ovaries homogenates stained with Coomassie (CBB), co-detection of haem-associated peroxidase activity with 3,3′-diaminobenzidine (DAB), and Western blot analyses of vitellogenin 1- and vitellogenin 2- cleavage products (α*Ir*Vg1 and α*Ir*Vg2, respectively). Note that the native *Ir*Vg1- and *Ir*Vg2-specific bands correspond to the positions of the major haemoproteins in tick ovaries (red asterisks). (C-D) RNAi-mediated silencing of *I. ricinus* vitellogenin 1 and 2. Unfed *I. ricinus* females were pre-injected with *gfp* dsRNA (control, *gfp*), *ir-vg1* dsRNA (*ir-vg1* KD), and *ir-vg2* dsRNA (*ir-vg2* KD), allowed to feed naturally on guinea pigs and then re-injected after detachment from the host with the same amount of dsRNA. (C) Effect of *I. ricinus* vitellogenin 1 and 2 RNAi-mediated silencing on ovaries appearance and haem levels. Bottom panels show the detailed view of ovary parts depicted by the yellow dashed squares above. Levels of haem *b* were determined by HPLC in three independent homogenates of ovaries dissected from each tick group 6 days after detachment. (D) Native pore-limit PAGE of ovaries homogenates (10 μg protein per lane) dissected 6 days AD from control (*gfp*), *ir-vg1* KD and *ir-vg2* KD ticks. Gels were stained with Coomassie (CBB) for proteins, 3,3′-diaminobenzidine for peroxidase activity of haem (DAB, red asterisks), and Western blot analyses were performed with antibodies against vitellogenin 1 (α*Ir*Vg1) and vitellogenin 2 (α*Ir*Vg2).

The following figure supplements are available for figure 6:

**Figure supplement 1.** SDS-PAGE and Western blot analyses of ovary homogenates from I. ricinus.

**Figure supplement 2.** Stage and tissue expression of *I. ricinus* vitellogenin 1 (*Ir*Vg1) and vitellogenin 2 (*Ir*Vg2).

**Figure supplement 3.** RNAi-mediated silencing of *I. ricinus* vitellogenin 1 and 2.

result points to a mutual regulation of both genes by an as yet unknown mechanism. More importantly, silencing of both *ir-vg1* and *ir-vg2* led to impaired development of tick ovaries and concomitant reduction in haem content in this tissue (*Figure 6C*). Additionally, native pore-limit PAGE followed by DAB-staining and Western blotting (*Figure 6D*) showed that decreased levels of both *Ir*Vg1 and *Ir*Vg2 in ovary homogenates from *ir-vg1* and *ir-vg2* KD ticks were associated with the disappearance of DAB-stained haem. These results collectively show that vitellins are the major haem-

binding proteins in *I. ricinus* ovaries and imply that the majority of haem is transported, along with vitellogenins, to the developing oocytes after tick detachment from the host.

## Discussion

The previous report on non-functional haem biosynthesis in the cattle tick *R. microplus* (*Braz et al., 1999*) prompted us to screen available mite genomes (namely: *I. scapularis, T. urticae*, and *M. occidentalis*) and reconstitute their gene repertoires for enzymes of the haem biosynthetic and degradative pathways. We found that 5-aminolevulinate synthase, together with the whole cytoplasmic segment of the haem biosynthetic pathway, is completely missing in hard ticks, but is present in other mites. Therefore we hypothesise that during evolution, ticks have lost most of the genes encoding haem biosynthesis as a consequence of their strict haematophagy.

Only three genes encoding the vestigial mitochondrial enzymes of the haem biosynthetic pathway, namely PPOX, CPOX and FECH, have been retained in the *I. scapularis* genome (*Gulia-Nuss et al., 2016*) (*Figure 1B*) and their orthologues were also identified in midgut and salivary gland transcriptomes of *I. ricinus* (*Kotsyfakis et al., 2015*) (*Figure 1—figure supplements 1–3*). PPOX transcripts were also found in salivary gland transcriptomes from various species of the genus *Amblyomma* (*Garcia et al., 2014*; *Karim and Ribeiro, 2015*). The same partial reduction in genomic coding for haem biosynthesis has been reported for a unicellular parasite, *Leishmania major*, in which the intracellular amastigote form expresses an active PPOX that likely sequesters the haem precursor coproporphyrinogen III from the macrophage cytosol to complete synthesis of its endogenous haem (*Zwerschke et al., 2014*). Another haem auxotroph, the nematode *Brugia malayi*, was suggested to bypass its incomplete haem biosynthetic pathway using tetrapyrrole intermediates from endosymbionts (*Wu et al., 2009*). Two lines of evidence suggest that PPOX, CPOX and FECH are not involved in haem biosynthesis in adult ticks: (i) Earlier, it was reported for *R. microplus* that no radioactively labelled δ-aminolevulinic acid was incorporated into haem present in haemolymph and ovaries (*Braz et al., 1999*); (ii) Recently, we have shown that RNAi-mediated silencing of the terminal FECH did not exert any effect on tick engorgement, oviposition, and larval hatching (*Hajdusek et al., 2016*). This data suggests that these remnants of the haem biosynthetic pathway in *I. ricinus* do not contribute to the tissue haem pool that sustains successful reproduction. Therefore the reason for retaining genes encoding the last three enzymes of the haem biosynthetic pathway in ticks remains obscure and should undergo further investigation.

The differential in vitro membrane feeding of *I. ricinus* females on whole blood (BF) or haemoglobin-free serum (SF) allowed us to investigate the importance of haemoglobin acquisition and intertissue transport of dietary haem in the hard tick *I. ricinus,* in an as yet unexplored way. These experiments surprisingly revealed that haemoglobin, which makes up about 70% of total blood proteins, is not a necessary source of amino acids for vitellogenesis (*Figure 2* and *Figure 3*). Moreover, we have unambiguously demonstrated that haem in tick eggs originates entirely from host haemoglobin acquired during female feeding on hosts. Serum-fed *I. ricinus* were capable of full engorgement and oviposition, however embryonal development and larval hatching was aborted (*Figure 2*). The capability of tick embryos to develop viable progeny could be fully rescued by addition as little as about 1% of the physiological concentration of haemoglobin (0.1% in serum) (*Figure 2—figure supplement 2*). In contrast to ticks, serum-fed triatomine *Rhodnius prolixus* were capable of laying eggs and giving rise to viable larvae (*Machado et al., 1998*). As *Triatominae* insects possess a complete haem biosynthetic pathway (*Kanehisa and Goto, 2000*), they can apparently reproduce even in the absence of dietary haem.

In the majority of animals studied so far (including insect blood-feeders), haem degradation represents the main source of iron, and conversely, iron is mainly utilised for *de novo* haem biosynthesis (*Zhou et al., 2007*; *Gozzelino and Soares, 2014*). Although it has been reported that, under certain conditions, haem can be degraded non-enzymatically (*Atamna and Ginsburg, 1995*), haem degradation-based on haem oxygenase (HO) is the most physiologically relevant (*Khan and Quigley, 2011*). We found that the HO gene was missing in the tick genome and correspondingly, the haem degradation product, biliverdin IX, could not be found in *I. ricinus* gut homogenates (*Figure 3—figure supplement 3*). We further noted that the absence of the HO gene is a common feature in other mite genomes (*Figure 1B*) and respective HO orthologues could not been found even in non-Acari genomes: the chelicerate genome of *Stegodyphus mimosarum* (*Sanggaard et al., 2014*) and the

myriapode genome of *Strigamia maritima* (*Chipman et al., 2014*). The apparent absence of HO transcripts in two color-polymorphic spiders of the genus *Theridion* is in agreement with the notion that these animals do not produce bilin pigments as haem degradation products (*Croucher et al., 2013*). As HO gene is present in the genomes of Hexapoda (*Adams et al., 2000*; *Holt et al., 2002*) and Crustacea (*Colbourne et al., 2011*), we hypothesise that the loss of HO is an old ancestral trait of Chelicerata and Myriapoda that are phylogenetically supported as sister groups (*Dunn et al., 2008*). Such a finding raises the question of dietary iron source for these animals, since iron is an essential electron donor/acceptor involved in vitally important physiological processes such as energy metabolism, DNA replication, and oxygen transport (*Hentze et al., 2004*; *Dunn et al., 2007*).

Earlier, we and others reported that successful tick development and reproduction is strictly dependent on the availability of iron and maintenance of its systemic homeostasis (*Hajdusek et al., 2009*; *Galay et al., 2013*). Here, we demonstrate that levels of intracellular ferritin, as an indicator of bioavailable iron, as well as the concentration of elemental iron, do not significantly differ in tick tissues dissected from BF and SF females (*Figure 3C,D*). These results further support the conclusion that bioavailable iron does not originate from haemoglobin-derived haem, but rather from serum iron-containing proteins, most likely host transferrin (*Hajdusek et al., 2009*; *Galay et al., 2014*; *Mori et al., 2014*). However, an unequivocal identification of the source(s) of bioavailable iron for tick metabolic demands has to await the implementation of a chemically defined artificial tick diet, as recently reported for the mosquito *Aedes aegypti* (*Talyuli et al., 2015*).

The entire dependence of ticks on haem derived from host haemoglobin underscores the importance of a deeper understanding of haem inter-tissue transport from the site of haemoglobin digestion in the gut to ovaries and other peripheral tissues. In the triatomine bug, *R. prolixus*, a 15-kDa haemolymphatic haem-binding protein (RHBP) was reported to transport haem to pericardial cells for detoxification and to growing oocytes for yolk granules as a source of haem for embryo development (*Walter-Nuno et al., 2013*). The haem transport and/or binding in ticks is mediated by HeLp/ CPs and vitellins (*Maya-Monteiro et al., 2000*; *Logullo et al., 2002*; *Boldbaatar et al., 2010*; *Smith and Kaufman, 2014*), that belong to the family of large lipid transfer proteins (LLTP) known to facilitate distribution of hydrophobic molecules across circulatory systems of vertebrates, as well as invertebrates (*Smolenaars et al., 2007*). Vitellogenins are reported to be expressed only in fertilised fully-fed females, whereas HeLp/CPs are expressed ubiquitously in various stages, including adult males, and tissues (*Donohue et al., 2008*; *Donohue et al., 2009*; *Khalil et al., 2011*; *Smith and Kaufman, 2014*). Based on these criteria, we clearly distinguished the *I. ricinus* carrier protein *Ir*CP3 from two vitellogenins, *Ir*Vg1 and *Ir*Vg2 (*Figure 5—figure supplement 1*; *Figure 6— figure supplement 2*) and demonstrated that during tick feeding, most haem in haemolymph is bound to *Ir*CP3. The haem is mainly transported to the developing ovaries during the off-host digestive phase, however the proportion of haem transported by *Ir*CP3 or vitellogenins remains to be investigated. In ovaries, haem is sequestered by vitellins serving as haem-storage proteins for embryonal development. Further studies of the native arrangement and haem-binding capabilities of tick vitellins are needed to determine whether one or both vitellin apoproteins are involved in haem binding.

Collectively, our results demonstrate that ticks lack functional haem biosynthesis, recycle dietary haem originating from digested haemoglobin, and the acquired haem does not contribute to the cellular iron pool. Therefore, haem and iron metabolism in ticks constitute a major departure from its canonical functioning described for other eukaryotic cells, where haem and iron homeostasis is based on balancing the flux between the opposing haem biosynthetic pathway and the HO-based degradative pathway. Further investigations of the exact molecular mechanisms involved in haem inter-tissue transport, intracellular trafficking, and compartmentation within the tick digest cells, promise to identify vulnerable targets in tick haem auxotrophy. This may lead to novel strategies for controlling ticks and the diseases that they transmit.

## Materials and methods

### Tick maintenance and natural feeding

A pathogen-free colony of *Ixodes ricinus* was kept at 24°C and 95% humidity under a15:9-hr day/ night regime. Twenty five females and males were placed into a rubber ring glued on the shaven

back of guinea pigs and ticks were allowed to feed naturally for a specified time or until full engorgement (7–9 days). Partially or fully engorged ticks were then either dissected or kept separately in glass vials until oviposition and larval hatching. All laboratory animals were treated in accordance with the Animal Protection Law of the Czech Republic No. 246/1992 Sb., ethics approval No. 095/2012.

## Tick membrane feeding in vitro

Membrane feeding of ticks in vitro was performed in feeding units manufactured according to the procedure developed by Kröber and Guerin (*Kröber and Guerin, 2007*). Whole bovine blood was collected in a local slaughter house, manually defibrinated and supplemented immediately with sterile glucose (0.2% w/vol). To obtain serum, whole blood samples were centrifuged at 2 500 × g, for 10 min at 4°C and the resulting supernatant was collected and centrifuged again at 10 000 × g, for 10 min at 4°C.

Diets were then supplemented with 1 mM adenosine triphosphate (ATP) and gentamicin (5 µg/ml), pipetted into the feeding units and regularly exchanged at intervals of 12 hr. For feeding, fifteen females were placed in the feeding unit lined with a thin (80–120 µm) silicone membrane, previously pre-treated with a bovine hair extract in dichloromethane (0.5 mg of low volatile lipids) as described (*Kröber and Guerin, 2007*). After 24 hr, unattached or dead females were removed and an equal number of males were added to the remaining attached females. For rescue experiments, pure bovine haemoglobin (Sigma, St. Louis, MO, H2500) was added to the serum diet since the 5th day of feeding at a concentration of 10%, 1%, or 0.1% (w/vol) and then feeding was resumed until tick full engorgement.

## Tissue dissection, haemolymph collection, and extraction of total RNA

Naturally or in vitro fed *I. ricinus* females were forcibly removed from the guinea pig or membrane at a specified time of feeding, or collected at a specified time after detachment. Haemolymph was collected into a glass capillary from the cut front leg, pooled, and used for subsequent experiments. Other tissues, namely ovaries, salivary glands, gut, tracheae with adjacent fat body cells, Malpighian tubules, and the remaining tissues tagged as 'rest' were dissected on a paraplast-filled Petri dish under a drop of DEPC-treated PBS. Total RNA was isolated from dissected tissues using a NucleoSpinRNA II kit (Macherey-Nagel, Germany) and stored at –80°C prior to cDNA synthesis. Total RNA from haemolymph was isolated using TRI reagent (Sigma). Single-stranded cDNA was reverse-transcribed from 0.5 µg of total RNA using the Transcriptor High-Fidelity cDNA Synthesis Kit (Roche Diagnostics, Germany). For subsequent applications, cDNA was diluted 20 times in nuclease-free water.

## Genome and transcriptome data mining

The search for tick genes encoding enzymes possibly involved in the haem biosynthetic and haem degradative pathways, a BLAST search using mosquito (*Anopheles gambiae*) genes was performed in the genome-wide database of *Ixodes scapularis* (https://www.vectorbase.org/organisms/ixodes-scapularis). Genes encoding canonical haemoproteins were identified based on their genomic annotation. Other mite genomes, namely *T. urticae* (*Grbić et al., 2011*) and *M. occidentalis*, were mined in available databases http://metazoa.ensembl.org/Tetranychus_urticae/Info/Index/ and http://www.ncbi.nlm.nih.gov/bioproject/62309, respectively. Additionally, transcriptomes available at the National Center for Biotechnology Information (http://www.ncbi.nlm.nih.gov) were screened using the BLAST® program. Metabolic pathways were reconstituted according to the Kyoto Encyclopedia of Genes and Genomes (*Kanehisa and Goto, 2000*).

## Expression and purification of recombinant proteins and production of antibodies

Gene products of 1806 bp, 2070 bp, 2151 bp, and 519 bp encoding fragments of *I. ricinus* carrier protein CP3 (*ir-cp3)*, *I. ricinus* vitellogenin 1 (*ir-vg1*), vitellogenin 2 (*ir-vg1*), and complete ferritin 1 (*ir-fer1*), respectively, were amplified from a whole body cDNA library using primers designed according to corresponding *I. scapularis* orthologues or the *ir-fer1* sequence (for primer sequences, see *Supplementary file 3*). Resulting amplicons were purified using the Gel and PCR Clean-up kit

(Macherey-Nagel), cloned into the pET100/D-TOPO vector of Champion pET directional TOPO expression kit (Invitrogen, Carlsbad, CA), and expressed using *E. coli* BL 21 Star (DE3) chemically competent cells. Expressed fusion proteins were purified from isolated inclusion bodies in the presence of 8M urea using a 5 ml HiTrap IMAC FF (GE Healthcare Bio-Sciences AB, Sweden) metal-chelating column charged with $Co^{2+}$ - ions and eluted with an imidazole gradient. The recombinant proteins (for sequences, see *Supplementary file 2*) were refolded by gradually decreasing the concentration of urea, finally dialyzed against 150 mM Tris/HCl, 150 mM NaCl, pH = 9.0, and used to immunize rabbits as described previously (*Grunclová et al., 2006*). The immune sera against *Ir*CP3, *Ir*Vg1, *Ir*Vg2 and *Ir*Fer1, tagged as α*Ir*CP3, α*Ir*Vg1, α*Ir*Vg2 and α*Ir*Fer1, were collected, aliquoted, and stored at –20°C until use.

## Tissue and developmental stage expression profiling by quantitative real-time PCR

cDNA preparations from developmental stages and tissues were made in independent triplicates and served as templates for the following quantitative expression analyses by quantitative real-time PCR (qPCR). Samples were analysed using a LightCycler 480 (Roche) and Fast Start Universal SYBR Green Master Kit (Roche). Each primer pair (for the list of qPCR primers, see *Supplementary file 3*) was inspected for its specificity using melting curve analysis. Relative expressions of *ir-cp3*, *ir-vg1* and *ir-vg2* were calculated using the ΔΔCt method (*Pfaffl, 2001*). The expression profiles from adult *I. ricinus* female tick tissues were normalized to *actin* and the developmental stage expression profiles were normalized to *elongation factor 1 (ef1)* (*Nijhof et al., 2009*; *Urbanová et al., 2014*).

## RNAi

A 521-bp fragment of *ir-cp3* (corresponding to positions 2688–3208 bp, GenBank KP663716), a 301-bp fragment of *ir-vg1* (corresponding to positions 2277–2577 bp of *I. scapularis* orthologue ISCW013727), a 303-bp fragment of *ir-vg2* (corresponding to positions 801–1103 bp of *I. scapularis* orthologue ISCW021228) were amplified from tick gut cDNA and cloned into the pll10 vector with two T7 promoters in reverse orientations (*Levashina et al., 2001*), using primer pairs CP3-F_RNAi, CP3-R_RNAi (*Supplementary file 3*) containing the additional restriction sites ApaI and XbaI. dsRNA of *ir-fer1* and *ir-irp* were synthesized as described (*Hajdusek et al., 2009*). Purified linear plasmids served as templates for RNA synthesis using the MEGAscript T7 transcription kit (Ambion, Lithuania). dsRNA (~1 µg in 350 nl) was injected into the haemocoel of unfed female ticks using Nanoinject II (Drummond Scientific Company, Broomall, PA). Control ticks were injected with the same volume of *gfp* dsRNA synthesized under the same conditions from linearized plasmid pll6 (*Levashina et al., 2001*). After 24 hr of rest in a humid chamber at room temperature, ticks were allowed to feed naturally on guinea pigs. The gene silencing was verified by qPCR and/or Western blot analyses.

## Reducing SDS-PAGE and Western blot

Tissue homogenates were prepared in 1% Triton X-100 in PBS supplemented with a Complete™ cocktail of protease inhibitors (Roche) using a 29G syringe, and subsequently subjected to three freeze/thaw cycles using liquid nitrogen. Proteins were then extracted for 1 hr at 4°C and 1 200 rpm using a Thermomixer comfort (Eppendorf, Germany). Samples were then centrifuged 15 000 × g, for 10 min at 4°C. Protein concentrations were determined using the Bradford assay (*Bradford, 1976*). Electrophoretic samples for SDS-PAGE were prepared in reducing Laemmli buffer supplemented with β-mercaptoethanol. Ten micrograms of protein were applied per lane unless otherwise specified. Proteins were separated on gradient (4–15%) Criterion TGX Stain-Free Precast gels (BioRad, Hercules, CA) in Tris-Glycine-SDS running buffer (25 mM Tris, 192 mM glycine, 0.1% (w/vol) SDS, pH 8.3) and visualized using TGX stain-free chemistry (BioRad). Proteins were transferred onto nitrocellulose using a Trans-Blot Turbo system (BioRad). For Western blot analyses, membranes were blocked in 3% (w/vol) non-fat skimmed milk in PBS with 0.05% Tween 20 (PBS-T), incubated in immune serum diluted in PBS-T (α*Ir*Fer1-1:50, α*Ir*Vg1-1:1 000, α*Ir*Vg2-1:1 000, α*Ir*CP3-1:1 000), and then in the goat anti-rabbit IgG-peroxidase antibody (Sigma) diluted in PBS-T (1:50 000). Signals were detected using ClarityWestern ECL substrate, visualized using a ChemiDoc MP imager, and analysed using Image Lab Software (BioRad).

Normalisation of Western blot analyses of gut homogenates were conducted using antibodies against *Ir*CP3, and homogenates of ovaries and eggs were normalised against the whole lane protein profile. Membrane stripping was carried out in a solution of 2% (w/vol) SDS and 0.5% (vol/vol) β-mercaptoethanol, and membranes were incubated for 1 hr at room temperature.

## Pore-limit native PAGE and detection of haem via peroxidase activity

Tissue homogenates were prepared as described above in Tris-Borate-EDTA (TBE) buffer (0.09M Tris, 0.08M boric acid, 2mM EDTA) supplemented with a Complete™ protease inhibitor cocktail (Roche). Electrophoretic samples for pore-limit native PAGE were supplemented with 10% (vol/vol) glycerol and 0.001% (w/vol) bromophenol blue. Samples were run in 4–16% Bis-Tris gel (Invitrogen) at 150 V in a cold room for 12 hr. Proteins were stained with Coomassie Brilliant Blue R-250 (CBB). For visualisation of haem-associated peroxidase activity, the gel was rinsed in water and then incubated in 100 mM sodium acetate pH 5.0 with 0.2% (w/vol) 3,3'-diaminobenzidine (DAB) and 0.05% (vol/vol) hydrogen peroxide (*McDonnel and Staehelin, 1981*). Alternatively, proteins were transferred onto nitrocellulose using a Trans-Blot Turbo system (BioRad) and used for Western blot analyses as described above.

## Light absorbance

Homogenates of five ovaries were prepared as described above in 400 µl TBE buffer and briefly spun down. Haemolymph samples were diluted 1:4 in TBE. Collected faeces (10 mg) were homogenised in 100 µl of TBE buffer and briefly spun down. Supernatants from all samples were applied in a 2 µl-drop on a NanoQuant Plate (Tecan, Austria) and absorbance over the UV-VIS spectrum was scanned using the model Infinity 200 M Pro microplate reader (Tecan).

## Haem *b* quantification

One dissected ovary, or 10 mg of eggs, was manually homogenised in methanol / 0.2% $NH_4OH$ (vol/vol) and centrifuged (15 000 × g, 10 min). The supernatant was discarded and haem was extracted from the pellet in 80% acetone / 2% HCl (vol/vol). The extract was immediately separated by HPLC on a Nova-Pak C18 column (4-µm particle size, 3.9 × 75 mm; Waters, Milford, MA) using a linear gradient of 25–100% (vol/vol) acetonitrile/0.1% trifluoroacetic acid at a flow rate of 1.0 ml/min at 40°C. Haem *b* was detected by a diode array detector (Agilent 1200; Agilent Technologies, Santa Clara, CA) and quantified using an authentic haemin standard (Sigma, H9039).

## Detection of biliverdin IX

Tick guts (wet weight ~20 mg) were dissected from naturally fed ticks 5 days after detachment from the guinea pig and homogenized individually in 100 µl of sterile PBS. For a positive control, 13 *Aedes aegypti* females were allowed to feed on mice and homogenized the 3[rd] day after feeding in 200 µl of sterile PBS. The samples were centrifuged (15, 000 × g, 10 min), supernatants were extracted in 80% acetone / 2% HCl (vol/vol) and separated by HPLC on a Zorbax Eclipse plus C18 column (3.5 µm particle size 4.6 x 100 mm, Agilent). A linear gradient (0–100%, 20 min) of solvent A (methanol: acetonitrile: 0.01 M sodium acetate pH 3.65; 1:1:2) and solvent B (acetonitrile / 0.1% TFA) at a flow rate of 0.6 ml/min at 40°C was used. Biliverdin IX and haem *b* were detected simultaneously using an Agilent 1200 diode array detector at wavelengths of 660 nm and 375 nm, respectively.

## Analysis of elemental iron

*I. ricinus* females were membrane fed on a blood or serum diet for 7 days until partial engorgement. Ovaries and salivary glands were dissected, taking special care to avoid contamination with gut contents, and washed in ultrapure 150 mM NaCl (TraceSELECT, Fluka, Switzerland). Pools of tissues, collected from 5 females, were spun down briefly to remove excess saline, and freeze-dried. The dry tissue samples were weighed on microbalances (with microgram precision) and submitted for elemental analysis using graphite furnace atomic absorption spectroscopy, kindly performed by Prof. Hendrik Küpper, Institute of Plant Molecular Biology, BC CAS, České Budějovice. The iron concentrations obtained were expressed in parts per million (ppm) related to the dry weight of tissues.

## Statistics

Data were analysed by GraphPad Prism 6 for Windows, version 6.04 and an unpaired Student's t-test was used for evaluation of statistical significance.

## Acknowledgements

This work was primarily supported by the Czech Science Foundation (GA CR) - grant No. 13-11043S to PK and, additionally by grant No. 14-33693S to DS, postdoctoral grants 13-27630P to OH and 13-12816P to RS. JP received support from the Grant Agency of the University of South Bohemia No. 074/2014/P. R.So. was supported by the National Programme of Sustainability (LO1416). The research at the Institute of Parasitology, BC CAS is covered by RVO 60077344. We thank Prof. Hendrik Küpper, Institute of Plant Molecular Biology, BC CAS for analysis of elemental iron by GF-AAS. We acknowledge the excellent technical assistance of Matěj Kučera, Lenka Grunclová, Jan Pilný, and Jan Erhart.

## Additional information

### Funding

| Funder | Grant reference number | Author |
| --- | --- | --- |
| Grantová Agentura České Republiky | 13-11043S | Petr Kopacek |
| Grant Agency of the University of South Bohemia | 074/2014/P | Jan Perner |
| National Program of Sustainability | LO1416 | Roman Sobotka |
| Grantová Agentura České Republiky | 13-12816P | Radek Sima |
| Grantová Agentura České Republiky | 14-33693S | Daniel Sojka |
| Grantová Agentura České Republiky | 13-27630P | Ondrej Hajdusek |

The funders had no role in study design, data collection and interpretation, or the decision to submit the work for publication.

### Author contributions

JP, Conception and design, Acquisition of data, Analysis and interpretation of data, Drafting or revising the article; RSo, RSi, JK, Participated in the experimental design, Approved the manuscript, Acquisition of data, Analysis and interpretation of data; DS, Participated in the experimental design, performed experiments, Approved the manuscript, Conception and design, Analysis and interpretation of data; PLdO, Analysis and interpretation of data, Drafting or revising the article; OH, Approved the manuscript, Conception and design, Acquisition of data, Analysis and interpretation of data; PK, Conception and design, Analysis and interpretation of data, Drafting or revising the article

### Ethics

Animal experimentation: All laboratory animals were treated in accordance with the Animal Protection Law of the Czech Republic No. 246/1992 Sb., ethics approval No.095/2012.

## Additional files

### Supplementary files

• Supplementary file 1. Prediction of genes coding for haemoproteins in the genome of *I. scapularis* and their putative function in tick metabolism.

• Supplementary file 2. Design, sequences, and sequence similarities of recombinant Vitellogenin_N domains of IrCP3, IrVg1, and IrVg2 used for raising specific antibodies.

• Supplementary file 3. Oligonucleotides used in this work.

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
