## [Decision Letter]

[Editors’ note: a previous version of this study was rejected after peer review, but the authors submitted for reconsideration. The first decision letter after peer review is shown below.]

Thank you for choosing to send your work entitled "Haem Acquisition and Distribution in Ticks" for consideration at *eLife*. Your full submission has been evaluated by Detlef Weigel (Senior editor) and three peer reviewers, one of whom is a guest of our Board of Reviewing Editors. The decision was reached after discussions between the reviewers. Based on our discussions and the individual reviews below, we regret to inform you that your work will not be considered further for publication in *eLife*.

While one cannot fail to recognize the large amount of time and effort that went into the work and reviewers appreciate attention to an important topic, a number of substantial concerns were raised. The major and unanimous consensus is that the study, as it currently stands, is premature and that a substantial series of additional experiments will be required to draw meaningful conclusions from the key questions. In particular, experiments concerning the dependency of larval viability solely on haem and non-utilization of haemoglobin haem as a source of iron, as well as roles and relationship of carrier protein and vitellogenin in haem transport or other aspects of tick physiology, require additional critical controls and more careful interpretation of the data. While the *I. scapularis* genome is incompletely annotated and data mining efforts possibly require careful extraction and analyses of trace files, there seems to be a missed opportunity in appropriate mapping of the orthologs in the haem biosynthetic pathway. The incomplete bioinformatics analyses thus precluded the formulation of tangible hypotheses, particularly one concerning the evolution of haem auxotrophy in ticks. Finally, the manuscript also needs improvement in textual organization, including with necessary details of certain result and methodological sections. Specific comments from the reviewers are appended below.

Reviewer #1:

The haem biosynthetic pathway remains highly conserved across metazoan evolution, although certain eukaryotes, including cattle ticks, were shown to be haem auxotrophs. The current manuscript by Perner et al. extends these observations by identifying haem auxotrophy in another major tick vector, Ixodes spp., that transmits a number of human infections. The approaches are ingenious, and the manuscript contains much important information regarding haem and iron homeostasis in an important disease vector. However, much of the data could be improved upon to present more compelling evidence to support the authors' major conclusions, which are noted below.

Major concerns:

1) Paragraph one, subheading “Host blood haemoglobin is expendable for tick feeding and oviposition but essential for reproduction”, Figure 2: What is truly interesting about this work is that haem seems to play a critical role in tick reproductive physiology. This speculation is derived from the key finding that unlike with blood, ticks fed on serum are unable to give rise to larvae. However, serum-fed ticks are not only deprived of hemoglobin from RBCs but also lack many other cellular/molecular components present in the blood. If haem is indeed the blood factor (absent in serum-fed ticks) that resulted in reproduction defects, then the authors could restore these defects with biochemical complementation by adding physiological concentrations of haem (or hemoglobin) to the serum used for tick feeding. This is a critical experiment that needs to be performed to support the authors' major conclusion.

2) Paragraph one, subheading “Haemoglobin is an indispensable source of haem, replaceable source of amino acids, and not a source of iron for ticks”, Figure 3: The proposition that haemoglobin is not a source of iron for ticks is another novel aspect of the study. However, the conclusion is based on indirect evidence that needs to be further substantiated. Iron concentration could be measured in tick tissues. The authors need to show that the amount of non-haem (serum) iron in blood-fed ticks is comparable to that of the total iron in serum-fed ticks.

3) The Method/Results section relevant to the in vitro membrane feeding system is severely deficient in details that would permit other labs to reproduce the work. The authors should mention the differences (in any) in feeding time, engorgement rates, percentage of egg-laying females in comparing blood with serum feeding. This would suggest how serum feeding impacts tick engorgement, growth, and embryogenesis.

4) The defect in larval hatching for serum-fed ticks is an interesting finding that also lack details and should be elaborated further. No evidence is provided other than an undefined image (Figure 2, bottom right panel) and a statement "[…]whereas no larvae hatched out of SF ticks' eggs". If these represent immature eggs, at least a thorough histological analysis would be useful.

Reviewer #2:

The main conclusions the authors draw are that 1) the tick *I. ricinus* is an obligate heme auxotroph and relies on hemoglobin from blood meals as a heme source; 2) Hemoglobin heme is not utilized as an iron source; 3) CP3 (HeLP) is the main hemoprotein in the hemolymph while vitellins bind heme in the ovaries. While the reliance on the hematocrit for larval viability is clear, it is not shown that this is specifically due to heme. While Hb heme may not be the preferred source of free-iron under normal conditions, mechanisms may exist to utilize heme as an iron source when iron becomes limiting. Finally, the requirement for CP3, Vg1, and Vg2 for inter-tissue heme transport or heme-dependent viability is poorly demonstrated, and maybe this is due to a lack of tractability of the tick model. In summary, new data are essential to support the major conclusions in this manuscript.

Major comments

1) While the authors show that blood cells are required for larval viability, it is not conclusively shown that this is heme-dependent. Exogenous heme can easily be added back into serum to establish whether hatching of progeny can be rescued, showing this phenotype is specifically due to a lack of heme and not due to another components of the hematocrit. Additionally, the efficiency of ferrochelatase knockdown is not shown. Protoporphyrin IX and iron can be added back to the serum to analyze whether the ferrochelatase is truly functional and lysates can be directly analyzed for FECH activity.

2) Genomic analyses and measuring ferritin levels is not sufficient to say the tick has no mechanism for removing iron from heme. There are examples of non-canonical heme oxygenases, for example in bacteria. To more definitively show that heme can utilized as an iron source, iron and heme-free serum should be supplemented with iron, heme, or both and then measurements of non-heme vs heme iron be performed in various tissues. Since HPLC data are already shown for heme levels, a direct measurement of heme degradation products / biliverdin can be made by HPLC. Changes in Ferritin levels are not a great indicator of iron levels, as mammalian Ftns are also acute phase proteins. Is Fer1/2 regulated by stress as by heme starvation?

3) The functional relationship between the homologs CP3, Vg1 and Vg2 is unclear. It is postulated that CP3 is the main heme carrier protein in the hemolymph but the reduction in heme content in eggs is minimal when CP3 is knocked-down, and it is not shown whether viable larvae arise from these eggs. If developing embryos are completely reliant on maternally-derived heme, there must be another mechanism for maternal heme delivery to ovaries. Does the *ricinus* genome code for additional CPs like scapulus? Additionally, if Vg1 and Vg2 bind heme after heme deposition in the ovaries, why would knockdown result in reduced heme in this tissue? What happens to intestinal heme content when CP3, Vg1 and Vg2 are knocked down? One might expect heme accumulation in the intestine or hemolymph if a heme carrier is depleted, depending on which tissue the carrier protein acquires heme.

Reviewer #3:

The paper "Haem Acquisition and Distribution in Ticks" by Perner et al. describes and functionally dissects heme metabolic pathway in tick *Ixoides scapularis* and *I. ricinus* using combination of genomic analysis and manipulative experiments. Authors develop an evolutionary scenario explaining heme metabolism that would support changes in *I. scapularis* and *I. ricinus* genome reflecting gene loss in heme synthesis and heme catabolism pathway.

Although manipulative experiments are elaborate and describe the transport of heme to the eggs, evolutionary scenario suffers from incompleteness.

First, it is known that the genome of *Ixoides scapularis* is not completely finished and is not annotated at the level representing the standard in the field.

Authors claim that they mapped orthologs of heme biosynthetic pathway in *I. scapularis*. However, in their Material and Methods section they do not describe how they performed their analysis. They claim that only two genes in heme biosynthetic pathway exist in *I. scapularis* genome, namely: "coproporphyrinogen oxidase [Vectorbase: ISCW010977] and ferrochelatase [Vectorbase: ISCW016187], were identified in the *I. scapularis* genome as shown in Figure 1." They state that other genes including aminolevulinic acid synthase, porphobilinogen synthase, hydroxymethylbilane synthase, uroporphyrinogen decarboxylase, uroporphyrinogen III synthase, uroporphyrinogen decarboxylase and protoporphyrinogen oxidase that are part of heme biosintethic pathway are not present in the genome stating: "Data mining of the available *I. scapularis* genome revealed the absence of six out of eight conserved enzymes involved in haem biosynthesis in most eukaryotes." However, when the reviewer examined orthologs of listed genes in heme biosynthetic pathway from related chelicerate genome (two spotted spider mite, *Tetranychus urticae*: http://bioinformatics.psb.ugent.be/orcae/overview/Tetur; Grbic et al. 2011), isolated first by blasting spider mite proteome with *Drosophila* heme synthesis ortologs, contrary to the claim of authors following *I. scapularis* possible ortologs were found in the genome:

1) aminolevulinic acid synthase (using *T.* urticae ortolog tetur32g00320):

ISCW020754-PA 268 E-value: 1e-88

Identities = 130/263 (49%), Positives = 180/263 (68%), Gaps = 2/263 (1%)

2) uroporphyrinogen decarboxylase (using *T. urticae* ortolog tetur19g03090)

ISCW020804-PA 215 E-value 3e-68

Identities = 125/336 (37%), Positives = 200/336 (60%), Gaps = 12/336 (4%)

3) protoporphyrinogen oxidase (using *T. urticae* ortolog tetur10g04900)

ISCW023396-PA protoporphyrinogen oxidase, putative|protein_cod… 55.3 1e-09

Identities = 36/105 (34%), Positives = 56/105 (53%), Gaps = 4/105 (4%)

Thus, 3 out of six genes claimed to be absent appear to be present in *I. scapularis* genome. Having in mind that all of listed genes were incompletely predicted and supported by few RNAseq (please compare *T. urticae* homologs) it cannot be excluded that the remaining 3 genes are present but in the portion of *I. scapularis* genome that has not been completed or not well annotated.

This should provide just a guidance and authors should confirm all these cases by proper alignment etc.

Also authors propose the scenario that loss of heme oxygenase represents specific feature of ticks as adaptation on heme-rich diet: "Therefore, we speculate that lack of haem oxygenase in ticks presents another unique feature of their adaptation to the haem-rich diet. This constitutes a major departure from the canonical functioning of haem and iron metabolism described for other eukaryotic cells, where haem and iron homeostasis is based on balancing flux between the opposing haem synthesis pathway and the haem oxygenase degradation pathway.

Authors conclude that "The lack of haem catabolism given the absence of haem oxygenase apparently presents another unique feature of tick adaptation to the haem-rich diet."

The quick analysis of *T. urticae* genome shows that this species also lack heme oxygenase. This species is plant-feeding and not exposed to heme-rich diet. Thus proposed evolutionary scenario is not sustainable. Possibility that it is simply missed in *T. urticae* is excluded by absence of it in related unpublished spider mite genomes.

Thus, an additional analysis and re-interpretation of data is necessary to support major conclusions of this paper.

[Editors’ note: what now follows is the decision letter after the authors submitted for further consideration.]

Thank you for resubmitting your work entitled "Acquisition of exogenous haem is essential for tick reproduction" for further consideration at *eLife*. Your revised article has been favorably evaluated by Detlef Weigel (Senior editor) and three reviewers, one of whom is a guest member of our Board of Reviewing Editors. The manuscript has been improved but there are some remaining issues that need to be addressed before acceptance, as outlined below:

The newly resubmitted manuscript addressed a number of important issues brought up during the original review. While all reviewers agreed that there is a substantial improvement in the quality of the data and their presentation, and that this is a fascinating study, a number of substantial concerns were raised, most of which require major editorial revision of the manuscript. Specific comments from the reviewers that require your attention are appended below.

1) Data presented in Figure 2 suggest critical involvement of hemoglobin in tick development. As the manuscript discuss heme auxotrophy, please clarify why reconstitution experiments involved supplementation of hemoglobin, rather than heme, to the SF media. Also clarify whether there is a correlation between numbers of live/hatched progeny with heme concentration. This is important because low heme may support hatching of larvae in the initial stages but may not be able to keep up as embryos utilize the limited supply of heme, as demonstrated by a previous article by Walter-Nuno et al., JBC 2013.

2) Perhaps the most interesting discovery here is that hemoglobin is required for embryo viability without serving as an iron or as an amino acid source. Then why is Hb required? Some additional experimentation could be conducted with their in vitro membrane feeding system, for example, adding other globular proteins or myoglobin to determine more specifically why Hb is required.

3) The proper way to determine whether a heme degradation system exists is to first deplete all sources of inorganic iron followed by supplementation of varying concentrations of heme as the sole iron source (Figure 3). Under these conditions, it is possible that ticks may be able to degrade heme to acquire iron. This result will indicate that heme degradation is conditional and induced when iron is limiting and that a non-canonical enzyme might be performing this function (as is found in several bacteria). Please clarify and discuss these possibilities.

4) The AAS result does not distinguish whether the iron was derived from Hb or other serum components (such as Tf). The result in Figure 3 is just suggestive, because SF conditions have the same iron and BF. Again, the only way to demonstrate this is to remove all sources of heme and inorganic iron followed by titrating heme and/or iron back.

5) Please clarify this statement "an efficient inter-tissue heme distribution system", when only 100 nmol of heme is being utilized from a blood meal to be transported to the ovaries. What was the method used to measure this?

6) The authors should discuss why the ticks have retained the last three enzymes in heme biosynthesis – could these enzymes serve another function? Could heme precursors from the host enter into this partial pathway? What is their expression over the course of tick feeding and development? Please refer to whether any of these genes (transcripts) were identified in published studies (especially recent RNA-Seq studies) involving Ixodes ticks.

7) The authors should provide some possibilities for where egg Fe is coming from if not Hb. They suggest other serum proteins such as transferrin. Could they test this by manipulating transferrin content in their in vitro assay?

8) The hypothesis stating that hemoglobin and serum proteins are endocytosed within gut cells via distinct mechanisms is not supported by solid experimental data, so please modify the statement.

9) Please clarify the statement "These results collectively show that vitellins are the major haemoproteins". Does vitellin actually function with heme bound, or is it a storage molecule for heme?

10) The authors say "these experiments revealed that haemoglobin was, surprisingly, not strictly required as a source of amino acids for vitellogenesis (Figure 2 and Figure 3)." Please clarify why this is surprising.

Finally, the discussion needs to be more cohesive and better link various results on tick metabolism and development into one complete story.

---

## [Author Response]

[Editors’ note: the author responses to the first round of peer review follow.]

*While one cannot fail to recognize the large amount of time and effort that went into the work and reviewers appreciate attention to an important topic, a number of substantial concerns were raised. […] Specific comments from the reviewers are appended below.*

Summary of the major changes, new data, and additional experiments carried out to improve the merit of our work presented in the new manuscript.

1) Data mining

In addition to the tick *Ixodes scapularis* genome, the genome-wide search has been performed in available genomes of other related mites namely the herbivorous mite *Tetranychus urticae* and the predatory mite *Metaseiulus occidentalis.* This mining, completed by a detailed phylogenetic analysis, confirmed that while non-hematophagous mites contain all genes coding for complete haem biosynthesis, the *I. scapularis* genome has retained only three genes encoding vestigial mitochondrial enzymes of haem biosynthesis, namely coproporphyrinogen-III oxidase (CPOX), protoporphyrinogen oxidase (PPOX) and ferrochelatase (FECH). Two other genes present in *I. scapularis* genome: δ- aminolevulinic acid synthase (ALAS) and uroporphyrinogen decarboxylase (UROD) are clearly of bacterial origin.

Another new discovery made by the genome wide searches was that gene coding for haem oxygenase is commonly missing also in other mites and possibly in chelicerates and myriapodes in general.

Therefore, we conclude that lack of haem oxygenase in ticks is not likely an evolutionary adaptation to the blood-feeding but rather an old ancestral trait of the last common ancestor of Chelicerata and Myriapoda.

2) Membrane feeding and rescue experiments

In-vitro membrane feeding of *I. ricinus* females was complemented by a rescue experiment performed by addition of different concentration of commercial bovine haemoglobin to the serum diet.

This conclusively confirmed that dietary haemoglobin is essential as a source of haem for embryonic development and tick reproduction. Full rescue of fertility is obtained with less than 1% of the physiological concentration of hemoglobin in blood.Under this condition, where haemoglobin contribution as an amino acid source is negligible, a significant amount of haem is still transferred to the eggs allowing normal embryo development. The embryonic development in tick eggs was examined by microscopy and the results are included in relevant figures.

3) The iron-level analysis in tick tissues

The analysis of elemental iron in tick peripheral tisssues was performed by a highly sensitive method of graphite-furnace atomic absorption spectroscopy (GF AAS) that allows determination of elemental iron concentration in a sub-milligram range of starting biological material (as is the case of pooled and dried tick tissues). Since GF AAS cannot distinguish between haem and non-haem iron, only ovaries and salivary glands dissected from partially engorged BF or SF ticks were analysed. The obtained results confirmed the previous data based on quantitative monitoring of ferritin 1 levels in tick tissues proving that haem is not a source of bioavailable iron for ticks.

4) Lack of haem oxygenase was confirmed by HPLC analysis of heam-derived pigments

In order to provide a stronger evidence that haem is not catabolised in ticks, we have adapted an HPLC method for detection of biliverdin IX in the tick gut homogenates. This analysis confirmed that no haem degradation product was present in the guts of fully engorged ticks, while we could detect biglutaminyl-biliverdin IX metabolite in homogenates from blood-fed mosquitoes, used as a positive control.

5) Removal of ferrochelatase functional assessment by RNAi

Data showing that RNAi-silencing of ferrochelatase had no effect on tick feeding success, oviposition, and reproduction, raised another questions about its physiological role, that we were not able to answer. As we felt that these data were contributing to a loss of focus on the main conclusions of this work, we used them in another manuscript with a distinct focus(Hajdusek et al. “Tick iron and heme metabolism -- new target for an anti-tick intervention”, currently under revision). We refer to these results in the Discussion with a tentative reference as (Hajdusek et al., submitted manuscript).

6) The manuscript was largely re-written to improve the overall readability, figures re-arranged to include new data, and the obtained results accordingly re-interpreted.

*Reviewer #1:*

*Major concerns:*

*1) Paragraph one, subheading “Host blood haemoglobin is expendable for tick feeding and oviposition but essential for reproduction”, Figure 2: What is truly interesting about this work is that haem seems to play a critical role in tick reproductive physiology. This speculation is derived from the key finding that unlike with blood, ticks fed on serum are unable to give rise to larvae. However, serum-fed ticks are not only deprived of hemoglobin from RBCs but also lack many other cellular/molecular components present in the blood. If haem is indeed the blood factor (absent in serum-fed ticks) that resulted in reproduction defects, then the authors could restore these defects with biochemical complementation by adding physiological concentrations of haem (or hemoglobin) to the serum used for tick feeding. This is a critical experiment that needs to be performed to support the authors' major conclusion.*

We agree with the reviewer that the lack of experiment with addition of haem or haemoglobin into the serum diet to rescue tick reproduction was a critical drawback. We performed this experiment with addition of physiological concentration (10%) of pure bovine haemoglobin and demonstrated that the embryonic development and larvae hatching was fully restored (see bottom panels of Figure 2). In addition, we performed this experiment with sub-physiological concentrations of haemoglobin (1% and 0.1%) and could demonstrate that as little as about one hundredth of haemoglobin levels in tick diet is sufficient to sustain their reproduction. We agree with the reviewer that supplementation the serum with haem (haemin) would be interesting. However, such an experiment requires a demanding optimization because of haem binding capacity of serum proteins (e.g. haemopexin, albumin).

Therefore simple addition of haemin into serum may lead to artefactual results which would be difficult to interpret.

*2) Paragraph one, subheading “Haemoglobin is an indispensable source of haem, replaceable source of amino acids, and not a source of iron for ticks”, Figure 3: The proposition that haemoglobin is not a source of iron for ticks is another novel aspect of the study. However, the conclusion is based on indirect evidence that needs to be further substantiated. Iron concentration could be measured in tick tissues. The authors need to show that the amount of non-haem (serum) iron in blood-fed ticks is comparable to that of the total iron in serum-fed ticks.*

We agree with the reviewer that the proof of bio-available iron content based on monitoring intracellular ferritin (Fer1) in tissues (such as used in our previous work, Hajdusek et al., PNAS, 2009) is indirect and requires substantiation. Therefore, we performed a determination of total elemental iron in tick tissues by GF AAS (see above, paragraph 3). This analysis confirmed that, apart from bioavailable iron, there is also no significant difference in levels of elemental iron in tissues from BF and SF ticks (Figure 3).

*3) The Method/Results section relevant to the in vitro membrane feeding system is severely deficient in details that would permit other labs to reproduce the work. The authors should mention the differences (in any) in feeding time, engorgement rates, percentage of egg-laying females in comparing blood with serum feeding. This would suggest how serum feeding impacts tick engorgement, growth, and embryogenesis.*

There were no obvious differences in the time-course of membrane feeding, feeding success, weights of engorged females and egg clutches sizes between BF and SF ticks. These experiments were repeated many times with reproducible results (see Figure 7 below).

Author response image 1.Table 1: Overview of the feeding and egg laying parameters of membrane-fed *I. ricinus* females.**DOI:**
http://dx.doi.org/10.7554/eLife.12318.026

*4) The defect in larval hatching for serum-fed ticks is an interesting finding that also lack details and should be elaborated further. No evidence is provided other than an undefined image (Figure 2, bottom right panel) and a statement "[…]whereas no larvae hatched out of SF ticks' eggs". If these represent immature eggs, at least a thorough histological analysis would be useful.*

We agree that our previous statement that “no larvae hatched out of eggs laid by SF ticks” was insufficient as a proof of impaired tick reproduction. Therefore, we performed a light microscopic examination of developing embryos in eggs laid by BF- and SF- ticks, as well as in the rescue experiment with different amount of haemoglobin in the serum diet (S+Hb ticks). We noted that embryonic development was blocked only in eggs from SF ticks (see Figure 2 and Figure 2—figure supplement 2).

*Reviewer #2:*

*Major comments*

*1) While the authors show that blood cells are required for larval viability, it is not conclusively shown that this is heme-dependent. Exogenous heme can easily be added back into serum to establish whether hatching of progeny can be rescued, showing this phenotype is specifically due to a lack of heme and not due to another components of the hematocrit. Additionally, the efficiency of ferrochelatase knockdown is not shown. Protoporphyrin IX and iron can be added back to the serum to analyze whether the ferrochelatase is truly functional and lysates can be directly analyzed for FECH activity.*

We appreciate the ideas of the reviewer how to proof the lack of haem biosynthesis in ticks. However, addition of just soluble haemin or haem precursor into the diet is a kind of oversimplification that does not take in account the complexity of tick digestive system. Haem is released from digested haemoglobin inside the digestive vesicles of the tick gut digest cells after its receptor-mediated endocytosis involving clathrin-coated pits. Therefore we used for the rescue experiment the pure commercial haemoglobin to demonstrate that haemoglobin alone (together with other serum components) and no other component of the red blood cells is required for successful tick reproduction. The consequence ‘no haemoglobin in the diet – no haem in eggs – no embryonic development’ is to our opinion a strong evidence that haem needed for tick reproduction originates exclusively from the host haemoglobin. Certainly, we cannot completely rule out that haem might be provided from other resources during the whole developmental cycle of the hard ticks, especially during the long inter-stage periods of starvation.

In the previous manuscript, the efficiency of ferrochelatase knockdown by RNAi was shown in the Figure 1—figure supplement 1, panel D. The information that ferrochelatase RNAi had no obvious effect on tick fecundity was included among screening of other genes/proteins possibly playing a role in tick haem and iron metabolism and was recently submitted to another journal. As mentioned above, this part was completely removed from the current version of the manuscript and we only refer to this result in the Discussion to support the conclusion that the retained haem biosynthetic genes do not play a role in the synthesis of endogenous haem in adult ticks.

*2) Genomic analyses and measuring ferritin levels is not sufficient to say the tick has no mechanism for removing iron from heme. There are examples of non-canonical heme oxygenases, for example in bacteria. To more definitively show that heme can utilized as an iron source, iron and heme-free serum should be supplemented with iron, heme, or both and then measurements of non-heme vs heme iron be performed in various tissues. Since HPLC data are already shown for heme levels, a direct measurement of heme degradation products / biliverdin can be made by HPLC. Changes in Ferritin levels are not a great indicator of iron levels, as mammalian Ftns are also acute phase proteins. Is Fer1/2 regulated by stress as by heme starvation?*

We agree with the reviewer that determination of bio-available iron by monitoring levels of intracellular ferritin 1 is only indirect proof of the lack of haem catabolism in ticks. Therefore, we performed additional experiments based on determination of elemental iron in tick tissues by GF-AAS (see above the paragraph 3 and response to the Reviewer #1, Major concern 2). Obtained results confirmed that iron levels in tick tissues are independent on the haemoglobin presence in tick diet.

Additional HPLC analysis of tick gut extract (see the paragraph 4 above) confirmed the absence haem- derived biliverdin in line with our previous hypothesis that lack of haem oxygenase in ticks is associated with the need to obtain iron from serum components and not from haem.

*3) The functional relationship between the homologs CP3, Vg1 and Vg2 is unclear. It is postulated that CP3 is the main heme carrier protein in the hemolymph but the reduction in heme content in eggs is minimal when CP3 is knocked-down, and it is not shown whether viable larvae arise from these eggs. If developing embryos are completely reliant on maternally-derived heme, there must be another mechanism for maternal heme delivery to ovaries. Does the ricinus genome code for additional CPs like scapulus? Additionally, if Vg1 and Vg2 bind heme after heme deposition in the ovaries, why would knockdown result in reduced heme in this tissue? What happens to intestinal heme content when CP3, Vg1 and Vg2 are knocked down? One might expect heme accumulation in the intestine or hemolymph if a heme carrier is depleted, depending on which tissue the carrier protein acquires heme.*

We agree with the reviewer that the relation between the haem-binding lipoproteins might seem unclear. In order to distinguish these two groups of lipoproteins, the knowledge of their stage and tissue expression profiles is needed. RNAi-mediated silencing of IrCP3 led to about 80% reduction of IrCP3 in tick haemolymph and only to about 50% reduction of haem in laid eggs. This can be explained either by the incomplete depletion of IrCP3 from tick haemolymph in KD ticks or by contribution of other CPs homologues possibly present in *I. ricinus* haemolymph. Alternatively, the haem transporting role is taken over by vitellogenins which presence in ovaries coincides with appearance of haem. Since we are aware about the uncertainty behind the haem inter-tissue transport in ticks, we interpret our obtained results in a more careful way and leave the ultimate answers to our future research. Here, we present only the unambiguous results: IrCP3 is the main haem-binding protein in *I. ricinus* haemolymph while vitellins are the main haem-binding protein in the ovaries.

*Reviewer #3:*

*Although manipulative experiments are elaborate and describe the transport of heme to the eggs, evolutionary scenario suffers from incompleteness.*

*First, it is known that the genome of Ixoides scapularis is not completely finished and is not annotated at the level representing the standard in the field.listed genes were incompletely predicted and supported by few RNAseq (please compare T. urticae homologs) it cannot be excluded that the remaining 3 genes are present but in the portion of I. scapularis genome that has not been completed or not well annotated.*

*This should provide just a guidance and authors should confirm all these cases by proper alignment etc.*

*[...] Thus, an additional analysis and re-interpretation of data is necessary to support major conclusions of this paper.*

We are grateful to the reviewer for his/her guide and advice regarding the mapping the haem biosynthetic pathway in *I. scapularis* using the orthologues from another mite, namely *T. urticae*. We agree that *I. scapularis* genome is still not perfectly annotated and contains a lot of gaps. In the previous manuscript, we only referred to the genes of haem biosynthetic pathway present in the KEGG database. Based on the reviewer’s suggestion, we performed a much more detailed data mining and BLAST analyses in available mite genomes as described in the paragraph 1 of this appendix.

Moreover, we confirmed transcripts of identified genes sequences by finding the corresponding orthologues in *I. ricinus* tissue transcriptomes. This led to additional identification of PPOX as a third gene of the tick haem biosynthetic pathway that was not revealed by KEGG. The reviewer found also other two genes related to *T. urticae* ALAS and UROD. Our phylogenetic analysis unambiguously revealed that these two genes are of bacterial origin.

A great hint from the reviewer #3 was the note about the lack of haem oxygenase (HO) in *T. urticae*. We further confirmed that absence of HO is a common feature among other Acari. This certainly makes indefensible our previous hypothesis that lack of HO presents another feature of tick adaptation to the haem-rich diet. In the light of a general absence of HO in Acari genomes, we may rather speculate that the inability to acquire iron from haem pushed the tick ancestor into haematophagy, and allowed the loss of haem synthesis as HO-mediated haem degradation is the major source of iron in most organisms studied up to now.

[Editors' note: the author responses to the re-review follow.]

*1) Data presented in Figure 2 suggest critical involvement of hemoglobin in tick development. As the manuscript discuss heme auxotrophy, please clarify why reconstitution experiments involved supplementation of hemoglobin, rather than heme, to the SF media. Also clarify whether there is a correlation between numbers of live/hatched progeny with heme concentration. This is important because low heme may support hatching of larvae in the initial stages but may not be able to keep up as embryos utilize the limited supply of heme, as demonstrated by a previous article by Walter-Nuno et al., JBC 2013.*

Supplementation of serum with haemin (soluble haem chloride) instead of haemoglobin (Hb) would be a more straightforward rescue experiment. However, addition of haemin into the diet is a non-physiological over-simplification that does not take into account the complexity of the tick digestive system. Unlike haematophagous insects, ticks digest blood meal intracellularly and haem is released from digested haemoglobin inside the digestive (lysosome-like) vesicles of the tick gut digest cells. This was the main reason why we performed the rescue experiment using pure commercial haemoglobin. Consequently, ‘no haemoglobin in the diet – no haem in the eggs – no embryonic development’ is, in our opinion, strong enough evidence that haem needed for tick reproduction originates exclusively from host haemoglobin and no other component of the host blood.

Irrespective of these considerations, we tried to verify what would happen if ticks were fed on serum supplemented with haemin, as suggested by the reviewer. Therefore, we performed the experiment as suggested by the reviewer. We prepared diets with 625µM (HH – high haemin) and 62.5µM (LH – low haemin) haemin in the serum, corresponding to a concentration of 1% and 0.1% w/v haemoglobin, respectively. Ticks were fed for the first five days on pure serum and from day 6, they received haemin-supplemented sera. Ticks were able to fully engorge and lay eggs under both conditions. Laid eggs were used for determination of haem concentrations by HPLC (Figure 8). Interestingly, haem concentrations were about the same in eggs from HH- and LH-fed females. We have no clear interpretation of this result. We assume that haemin uptake might be facilitated through haem-binding serum proteins (e.g. albumin, haemopexin). Similar haem concentrations in egg clutches of HH and LH ticks may be attributed to limited haem-binding capacity of serum proteins.

A quantitative correlation between haemoglobin (haem) concentration in the diet and the number of hatched living larvae is quite difficult to determine. We routinely estimate the larval hatching rate by crosses +, ++, and +++ for low, medium, and high numbers, respectively. There is great variability in larval numbers, even for naturally fed ticks, and we further noted that there are also seasonal variations in clutch sizes and the time needed for larval hatching. Therefore, it is not feasible to reliably evaluate differences in hatching from in vitro fed females in relation to haemoglobin concentrations over a reasonable time frame.

In the case of the triatominae bug *Rhodnius prolixus* mentioned by the reviewer, RNAi-mediated silencing of haemolymphatic haem-binding protein (RHBP) resulted in the laying of viable red eggs and non-viable white eggs in earlier and later stages of oviposition, respectively (Walter-Nuno et al., 2013). In contrast, ticks fed on different haemoglobin/haemin concentrations laid fairly homogenously colored clutches of eggs (Figure 2, Figure 2—figure supplement 2 and Figure 8). To this end, we did not observe any obvious difference in larval hatching from individual parts of any egg clutch.

*2) Perhaps the most interesting discovery here is that hemoglobin is required for embryo viability without serving as an iron or as an amino acid source. Then why is Hb required? Some additional experimentation could be conducted with their in vitro membrane feeding system, for example, adding other globular proteins or myoglobin to determine more specifically why Hb is required.*

We believe that we have sufficiently demonstrated in our manuscript that haemoglobin is strictly required only as a source of haem needed as the prosthetic group for endogenous haemoproteins (see [Supplementary-material SD1-data]). Nevertheless, we thank the reviewer for his/her proposal to perform an additional experiment to replace haemoglobin with myoglobin in the diet. Despite this non-physiological situation, it appears that ticks, indeed, are capable of acquiring haem from myoglobin and can transport it to the developing oocytes (Figure 8). As in the case of haemin, it is not clear whether the uptake of myoglobin by tick digestive cells follows the proposed haemoglobin pathway via the putative specific receptor-mediated endocytosis or is absorbed non-specifically by fluid-phase pinocytosis proposed for serum proteins. Further studies are planned to answer this question.

Author response image 2.Experimental feeding in vitro of *I.* ricinus females on serum supplemented with myoglobin or haemin.Ticks were membrane fed in vitro on bovine serum (S). Pure equine myoglobin (Sigma, M0630) or haemin (Sigma, H9039) of specified concentrations were added to the serum diet from the 6th day of feeding and feeding was then resumed until tick full engorgement. The fully engorged females were weighed, allowed to lay eggs, and haem concentrations in eggs were determined by HPLC. (**A**) – representative females fed on respective diets laying eggs; note the different female coloration due to distinct amounts of haem in the diet, yet egg clutches are similarly coloured. (**B**) Weights of fully engorged females fed on respective diets; each symbol presents the weight of one fully engorged female; bar charts depict the mean ± SEM. (**C**) Levels of haem b were determined by HPLC in egg homogenates from ticks fed on sera supplemented with 1% w/v myoglobin, 625 μM haemin, 62.5 μM haemin, or pure serum. Data (mean values ± SEM) were acquired from homogenates of three independent clutches of eggs. For further details see the Material and Method section.**DOI:**
http://dx.doi.org/10.7554/eLife.12318.027

*3) The proper way to determine whether a heme degradation system exists is to first deplete all sources of inorganic iron followed by supplementation of varying concentrations of heme as the sole iron source (Figure 3). Under these conditions, it is possible that ticks may be able to degrade heme to acquire iron. This result will indicate that heme degradation is conditional and induced when iron is limiting and that a non-canonical enzyme might be performing this function (as is found in several bacteria). Please clarify and discuss these possibilities.*

This is definitely a correct suggestion, but unfortunately unfeasible under the current state-of-the-art of tick artificial feeding. Our efforts to manipulate the serum diet usually resulted in a failure of tick feeding. For instance, ticks do not feed on serum that has been dialyzed (removal of low molecular weight components). On the other hand, ticks can be partially fed on a serum ultrafiltrate (< 3 kDa), however they do not commence the rapid engorgement (‘big sip‘) phase in the absence of serum proteins. Hence, we have so far failed to establish conditions of limited iron supply in the tick diet. Currently, implementation of a defined artificial tick diet, similar to that found for *Aedes aegypti* (Talyuli et al., 2015), seems to be the only way to experimentally approach the question of the source of iron and other essential nutritional components.

*4) The AAS result does not distinguish whether the iron was derived from Hb or other serum components (such as Tf). The result in Figure 3 is just suggestive, because SF conditions have the same iron and BF. Again, the only way to demonstrate this is to remove all sources of heme and inorganic iron followed by titrating heme and/or iron back.*

We believe that this comment probably resulted from our unclear explanation of the experiment shown in the Figure 3. Exactly because AAS is not able to distinguish between haem and non-haem iron, we chose to measure total iron in peripheral tissues (ovaries and salivary glands) dissected from semi-engorged females where still no haem is transferred to ovaries. If iron transported to these peripheral tissues by secreted ferritin 2 (Hajdusek et al., 2009) originated from haem degradation in guts, one should expect that the amount of iron in tissues from BF ticks would be much higher than found in SF ticks. Our AAS analysis therefore, excluded this hypothesis, providing additional evidence that haem is not a source of bioavailable iron in ticks. We have modified the corresponding sentence in Results to better clarify the rationale of this experiment:

“As this method is not able to distinguish between iron of haem and non-haem origin, only salivary glands and ovaries dissected from partially engorged BF and SF ticks were used for the analysis to avoid distortions caused by the presence of haemoglobin in the samples”.

*5) Please clarify this statement "an efficient inter-tissue heme distribution system", when only 100 nmol of heme is being utilized from a blood meal to be transported to the ovaries. What was the method used to measure this?*

We thank the reviewer for pointing out the obvious contradiction in this sentence. We have re-phrased it accordingly:

“We estimate that out of approximately 10 µmol of total haem acquired from a tick blood meal, only about 100 nmol (~1%) needs to be transported to the ovaries within a period of several days[…]”

Our estimation of the quantity of total haem acquired from a blood meal (~10 µmol) was based on the following calculation: A fully engorged female imbibes approximately 1 ml of blood meal containing ~150 mg haemoglobin (2.32 µmol) and each haemoglobin molecule contains four haem rings (~10 µmol). The average haem content in eggs from ticks fed on whole blood was approximately 600 pmol/mg, which, when multiplied by the weight of a typical egg clutch (100–150 mg) yields 100 nmol of haem.

*6) The authors should discuss why the ticks have retained the last three enzymes in heme biosynthesis – could these enzymes serve another function? Could heme precursors from the host enter into this partial pathway? What is their expression over the course of tick feeding and development? Please refer to whether any of these genes (transcripts) were identified in published studies (especially recent RNA-Seq studies) involving Ixodes ticks.*

We have examined the available RNA-Seq data from *I. ricinus* salivary glands and midgut transcriptomes (Kotsyfakis et al., 2015) and found that *ir-cpox, ir-ppox* and *ir-fech* are indeed expressed in this species (Figure 9). To obtain a deeper insight into putative functions of encoded proteins, we have carried out expression profiling of *ir-cpox, ir-ppox*, and *ir-fech* over tick active developmental stages (larvae, nymphs, adult males and females) (Figure 9) and tissues dissected from partially-engorged females (Figure 9). Genes *ir-cpox* and *ir-ppox* showed the highest expression in unfed larvae and in ovaries from partially-engorged females, while the *ir-fech* transcript was detected in all cDNA sets examined. As all three genes have retained mitochondrial target sequences (based on TargetP prediction), we speculate that the encoded proteins might have adopted certain functions in mitochondrial biology, possibly distinct from haem biosynthesis. Similar qPCR expression profiles of *ir-cpox* and *ir-ppox* suggest that at least two encoded proteins (CPOX and PPOX) may be part of the complex metabolon (Medlock et al., PLoS One, 2015).

Author response image 3.Expression of coproporphyrinogen-III oxidase (ir-cpox), protoporphyrinogen oxidase (ir-ppox), and ferrochelatase (ir-fech) in the tick *Ixodes ricinus*.(**A**) Expression profiles of nymphal and female adult *I. ricinus* stages over initial phases of feeding from available RNAseq data (Kotsyfakis et al., 2015). SG – salivary glands; MG – gut; N – nymph; A – adult; 12, 24, 36 – 12, 24, 36 hours of feeding. (**B**) qPCR analyses of *ir-cpox, ir-ppox*, and *ir-fech* expression profiles over active developmental stages of *I. ricinus* and (**C**) over tissues dissected from partially-engorged females fed for 6 days (**C**). Data were obtained from three independent cDNA sets, and normalized to elongation factor 1α (*ef1α*). UF – unfed; FE – fully engorged; SG – salivary glands, OVA – ovaries; Trachea – trachea-fat body complex; MT – Malpighian tubules; Rest – remaining tissues. Mean values +/- SEM are shown.**DOI:**
http://dx.doi.org/10.7554/eLife.12318.028

*7) The authors should provide some possibilities for where egg Fe is coming from if not Hb. They suggest other serum proteins such as transferrin. Could they test this by manipulating transferrin content in their in vitro assay?*

As already mentioned in our response to point 2, we could not find any way to deplete transferrin from bovine serum without the manipulation leading to a failure of tick feeding. Instead, we increased the amount of transferrin in bovine serum by addition of 3 mg/ml of commercially available holo-transferrin. Addition of this iron-saturated transferrin to the serum diet led to a corresponding increase in intestinal ferritin 1 used for monitoring the levels of bioavailable iron, as described earlier (Hajdusek et al., 2009) (Figure 10). This experiment indicates that ticks are indeed capable of acquiring iron from host transferrin. However, it is still not clear whether the host transferrin is the exclusive source of bioavailable iron for ticks. Answering this question on the source of iron for ticks lacking the haem oxygenase gene is a challenging task for our future research, possibly based on using defined artificial diets as discussed in our response to point 3.

Author response image 4.Increased transferrin levels in serum led to higher levels of ferritin 1 in tick gut.Ticks were fed on serum or on serum supplemented with 3 mg/ml of bovine holo-Transferrin (Sigma, T1283). This addition increased the concentration of transferrin in serum approximately 2-fold, whereas the amount of transferrin iron was increased 3‒4 fold (iron saturation of natural transferrin in serum is usually about 30%). (**A**) SDS PAGE of diets (10 µg per lane) used for the experiment: S – serum; S+Tf – serum with added 3 mg/ml of iron-saturated transferrin, stained with Coomassie blue (CBB) and Western blot with anti-transferrin specific antibodies (αTf). (**B**) SDS PAGE of gut homogenates dissected on the 4th day of feeding (Day 4) and from fully engorged females (Day 8) visualized using the TGX Stain-Free technology (TGX) and Western blot detection of tick ferritin 1 levels using specific antibodies against recombinant *I. ricinus* ferritin1 (αIrFer1).**DOI:**
http://dx.doi.org/10.7554/eLife.12318.029

*8) The hypothesis in L72-274 stating that hemoglobin and serum proteins are endocytosed within gut cells via distinct mechanisms is not supported by solid experimental data, so please modify the statement.*

The reviewer is correct in saying that the concept of separate haemoglobin uptake via receptor-mediated endocytosis involving clathrin-coated pits and fluid-phase endocytosis for serum proteins (e.g. albumin) has not yet been unambiguously proved. Also the haemoglobin receptor expected to be present on tick digest cells has not yet been identified. Due to the uncertainty in this matter, the corresponding part of the text was removed from the Discussion.

*9) Please clarify the statement "These results collectively show that vitellins are the major haemoproteins". Does vitellin actually function with heme bound, or is it a storage molecule for heme?*

We agree with the reviewer that vitellins are not haemoproteins in the strict sense as they apparently do not need haem as a prosthetic group for their function. The same is true for HeLp, the haemo-lipoprotein in tick hemolymph *Ir*CP3. Given their lipophilic character, these low-density lipoproteins function as haem scavengers, and most-likely as transporters and storage proteins for insoluble haem. Based on this, we replaced the term ‘haemoprotein’ by the term ‘haem-binding proteins’ where appropriate.

*10) The authors say "these experiments revealed that haemoglobin was, surprisingly, not strictly required as a source of amino acids for vitellogenesis (Figure 2 and Figure 3)." Please clarify why this is surprising.*

Based on the fact that haemoglobin makes up about 70% of total blood proteins, it has long been assumed that haemoglobin was an indispensable source of amino acids for the production of yolk proteins. For this reason, our research over the past decade was focused mainly on haemoglobin digestion as a potential target for anti-tick intervention. However, the results presented in this work show that ticks can produce an equal amount of vitellogenins and eggs even in the absence of haemoglobin. In order to make this surprising result more obvious we changed the corresponding sentence in the following way (see lines 264‒266) ‘These experiments surprisingly revealed that haemoglobin, which makes up about 70% of total blood proteins, is not a necessary source of amino acids for vitellogenesis (Figure 2 and Figure 3)”.

*Finally, the discussion needs to be more cohesive and better link various results on tick metabolism and development into one complete story.*

We thank the reviewers for this helpful suggestion. We have removed some parts that were not tightly associated with our results and that possibly would be more appropriate to be discussed in a review article concerning other haem auxotrophic organisms. We believe that the discussion is now more focused on blood-feeding arthropods without losing general interest for other biologists.